# Does co-inoculation of mycorrhiza and *Piriformospora indica* fungi enhance the efficiency of chlorophyll fluorescence and essential oil composition in peppermint under irrigation with saline water from the Caspian Sea?

Masoumeh Khalvandi 📕[1] *, Mohammadreza Amerian[1], Hematollah Pirdashti[2], Sara Keramati[2]

1 Department of Agronomy, Faculty of Agriculture, Shahrood University of Technology, Shahrood, Iran,
2 Department of Agronomy, Genetic and Agricultural Biotechnology Institute of Tabarestan, Sari Agricultural Sciences and Natural Resources University, Sari, Iran

* M.khalvandi@gmail.com

## Abstract

Symbiotic associations with endophytic fungi are ecologically important for medicinal and aromatic plants. Endophytic fungi highly affect the quantity and quality of herbal products. In this study, a pot experiment was carried out in the greenhouse to investigate the interactive effects of *Piriformospora indica* and arbuscular mycorrhizal (AMF) inoculation on the chlorophyll fluorescence, essential oil composition, and antioxidant enzymes of peppermint under saline condition. The results showed that Fo, YNPQ, YNO, and NPQ values were obviously increased under salinity conditions, while essential oil content, chlorophyll a and b, gs, Fm, Fv, ETR, ΦPSII and Fv/Fm ratio decreased by increasing salinity. In addition, salt induced the excess $Na^+$ uptake, whereas the opposite trend was observed for P and $K^+$. The synergistic association of *P. indica* and AMF caused a considerable increase in the antioxidant ability, essential oil content, Fv/Fm ratio, ΦPSII, and amount of P and $K^+$ uptake in salt-stressed plants. The main peppermint oil constituents, menthol, menthone, and 1,8-cineole increased considerably in inoculated plants. Besides, the applied endophytic fungi positively enhanced the ability of peppermint to alleviate the negative effect of the salinity stress.

## 1. Introduction

*Mentha piperita* is one of the most important medicinal plants of Lamiaceae family, the demand of which has grown dramatically in recent decades. The essential oil of peppermint is widely used in food, fragrance, pharmaceutical and cosmetic industries [1, 2]. It is evident that the essential oil contains a high proportion of cytotoxic, antioxidant, and anti-microbial compounds with health-beneficial effects such as antifungal effects, cancer prevention activities as well as scavenging of reactive oxygen species (ROS) activity because of its valuable constituents

**Data Availability Statement:** All relevant data are within the manuscript and its Supporting Information files.

**Funding:** There was no funding for this article. The funders had no role in study design, data collection and analysis, decision to publish, or preparation of the manuscript.

**Competing interests:** The authors have declared that no competing interests exist.

[1, 3, 4]. The sustainability of medicinal plants' yield is negatively affected by salt stress in many parts of the world [5]. Salinity causes a broad range of physiological changes in plants, induces impairments of metabolic processes, such as photosynthesis, pigment synthesis, secondary metabolite accumulation, destruction of the chloroplast and thylakoid systems, and inhibits the photosystem II (PSII) activity, which results in the excess accumulation of ROS [6, 7]. In this regard, chlorophyll fluorescence is widely used to detect plant physiological status and to determine photosynthetic damage under various stresses, because the same can show the integrity of the thylakoid membrane, the quantum yield of photosystem II (PSII), and the balance between the metabolism process and energy production [8, 9]. It has been proved that salinity has a negative effect on the photochemical efficiency of PSII as shown by chlorophyll fluorescence parameters such as qP, ΦPSII, and Fv/Fm [10, 11]. In this regard, previous research showed that salt stress reduced the quantum yield of photosystem II; whereas, it increased non-photochemical extinction in lettuce plants [10], and decreased the ratio of the quantum yield of actual PSII photochemistry in durum wheat [12]. Monoterpenes play an important role in plants' response to environmental stresses. Nevertheless, there are few studies focusing on the relationship between terpene biosynthesis and the efficiency of PSII. Terpenoid biosynthesis patterns are mostly affected by photosynthetic carbon assimilation and partly regulated by environmental stresses [13, 14]. There are also some reports about the anti-stress activity of monoterpenes in plants under stressful conditions. For example, it has been reported that the Monoterpenes in *Quercus ilex* can improve membrane integrity by ROS-scavenging, photoprotective, and thermotolerance roles [15, 16].

Applying plant beneficial rhizospheric microorganisms (PBRMs) like symbiotic fungi is an important environmental strategy to overcome the deleterious effects of salinity stress, and to improve plants' yield and performance. Arbuscular mycorrhizal fungi and *Piriformospora indica* (also named as *Sebacinales indica*) are two of these beneficial endophytic fungi which increase plant resistance to various environmental stresses through a wide range of mechanisms; mechanisms such as stimulating the immune system of plants [17, 18], promoting nutrient uptake [19], and stimulating root growth [20]. It is believed that symbiotic relationship with *P. indica* can help to ameliorate photo-oxidative damages to PSII and electron transfer chain [21], also, it increases the synthesis of secondary metabolites in various plants [22, 23]. Several studies have shown that symbiotic relationship with *P. indica* can stimulate the human health-promoting compounds synthesis in various plants. For example: podophyllotoxins in *Linum album*, thymol and carvacrol in *Thymus vulgaris*, menthol in *Mentha piperita* [24], and estragole in *Ocimum basilicum*, pharmaceutically useful compounds [25, 26]. Moreover, it can reduce undesirable compounds for human nutrition, such as erucic acid and glycosylates contents in *Brassica napus* [27].

The present study aims to investigate the hypothesis that co-inoculation of AMF and *P. indica* can protect peppermint plants against adverse effects of the salt stress through modifying some physio-biochemical characteristics.

## 2. Materials and methods

### 2.1. Growth conditions

Plants were grown in the greenhouse at Genetics and Agricultural Biotechnology Institute of Tabarestan at Sari Agricultural Sciences and Natural Resources University. A factorial experiment was carried out in a completely randomized design with three replications. Plants were treated with four levels of fungal inoculation including no-inoculation (control), *Piriformospora indica* (*P. indica*), Arbuscular mycorrhizal fungi (AMF), and co-inoculation with *P. indica* and AMF (*P. indica* + AMF) under four salinity levels including 0, 3, 6 and 9 dSm$^{-1}$ (the Caspian Sea water and distilled water mixture was used for irrigation) (Table 1). In order to

**Table 1. Chemical analysis of Caspian Sea water [18].**

| parameter | unit | amount |
|---|---|---|
| EC | dS m$^{-1}$ | 15 |
| pH | - | 8.2 |
| sodium | ppm | 376.48 |
| potassium | ppm | 77.11 |
| calcium | ppm | 17 |
| magnesium | ppm | 54.5 |

reach each level of salinity stress: determined amount of the Caspian Sea water and distilled water were mixed and then the desired salinity was determined by EC meter. Mycorrhiza fungi inoculum (consisted of spores in a sand and mycorrhizal roots mixture) was prepared from Turan Biotechnology Company, Shahrood, Iran. The *Piriformospora indica* culture was kindly gifted by Prof. Karl-Heinz Kogel, Institute of Phytopathology and Applied Zoology, University of Giessen, Germany. *P. indica* was cultured in liquid Kafer's medium at 24°C for 10 days [28]. 10 g of Arbuscular mycorrhizal inoculum (with a density of 120 active spores per gram) and 10 ml of *Piriformospora indica* suspension ($1\times 10^9$) were added to the pots. Peppermint rhizomes were planted in plastic pots, and were irrigated with saline water after four weeks of planting. Plant physiological measurements and sampling for the desired traits were carried out after two months of planting in all plants. The roots were bleached in a solution of KOH 10%, they were painted, using 5% solution of ink and vinegar. Then, 40 pieces of stained root were spread out in a Petri dish. The colonization percentage was measured by using the gridline intersects method optical microscope (10- 40X) [29–31].

## 2.2. Physiological parameters

Chlorophyll fluorescence parameters were measured in the last fully developed leaf by using pulse amplitude modulated fluorometer (PAM-2500, Walz, Germany). First, leaves were placed in darkness for 30 minutes using specific leaf clamps. The samples were exposed to low-intensity light [< 0.1 μmol (photon) m−2 s−1, red light]. Then, a saturating light pulse [> 8,000 μmol (photon) m−2 s−1, white light) was turned on for 1 s (one pulse). The minimum fluorescence (Fo) and maximum fluorescence (Fm) were determined in dark-adapted leaves. The variable fluorescence (Fv) and maximum quantum photosystem II efficiency (Fv/Fm) were evaluated based on Eqs 1 and 2. and the effective photochemical quantum efficiency II [ΦPSII], electron transfer rate [ETR, μmol(electron) m–2 s–1], non-photochemical quenching (NPQ), the quantum yield of regulated energy dissipation (YNPQ), and Quantum yield of non-regulated energy dissipation in PSII (Y (NO)) were calculated by using Eqs 3 and 6 [12].

$$Fv = Fm - Fo \tag{1}$$

$$Fv/Fm = (Fm - Fo)/Fm \tag{2}$$

$$\Phi PSII = (Fm' - F)/Fm' \tag{3}$$

$$NPQ = (Fm/Fm') - 1 \tag{4}$$

$$Y(NPQ) = (F/Fm') - (F/Fm) \tag{5}$$

$$Y(NO) = F/Fm \qquad \qquad 6)$$

F = steady-state fluorescenc

Fm′ = maximum fluorescence measured in light-exposed leaf samples.

Stomatal conductance (gs) was determined by using a porometer data. Chlorophyll a and b concentrations were calculated following Hameed et al. [32].

The minerals $Na^+$ and $K^+$, P in peppermint leaves were determined according to Ntatsi et al. [33]. The amount of $K^+$, P and $Na^+$ were assayed via an atomic absorption spectrophotometer, microplate spectrophotometer, and flame photometer, respectively.

Membrane permeability was monitored based on procedures described in Lutts et al. [34].

Essential oil extraction:

The essential oil of peppermint was extracted from dried leaves by using hydro-distillation in Clevenger's apparatus. Chemical composition of peppermint essential oil (Fig 1) was analyzed by gas chromatography/mass spectrometry (GC/MS) [18].

## 2.3. Statistical analysis

SAS (9.2) statistical program was used for statistical analysis and a mean between treatments was compared using an LSD (Least Significant Difference) test (P <0.05). Principal component analysis (PCA) and Heatmap was performed with the R language.

## 3. Results

Analysis of variance showed that salinity stress, *P. indica*, and AMF significantly affected all measured traits. According to the results, a significant two-way interaction was observed between salinity and inoculation treatment in essential oil, Na, membrane electrolyte leakage, Fv/Fm, ΦPSII, Fo, and stomatal conductance (Tables 2 and 3); however, no significant Interaction was observed in other parameters. The main effects were significant for Fm, Fv, P, K, ETR, NPQ, Y(NPQ), Y(NO), chlorophyll a, and chlorophyll b (Tables 2 and 3).

## 3.1. Colonization rate

In the microscopic investigation, not only chlamydospores and hyphae of *P. indica*, and vesicles, arbuscules, and intraradical hyphae of AMF were observed in the peppermint root cortex,

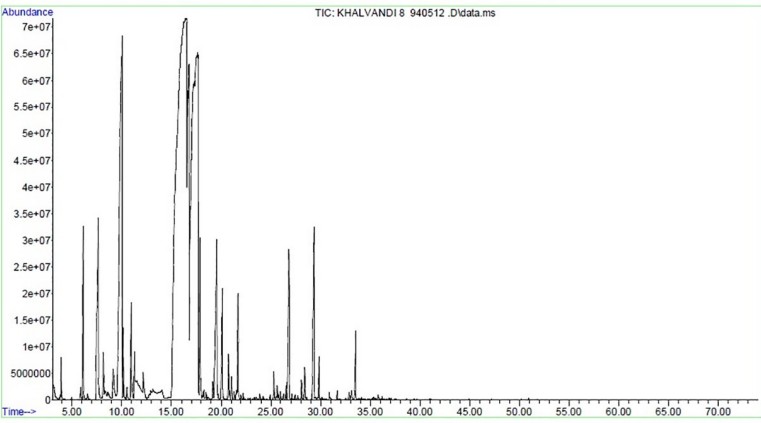

**Fig 1. *Mentha piperita* essential oil chromatogram.**

**Table 2. Variance analysis of salinity and *P. indica* and AMF effects on measured chlorophyll fluorescence parameters of peppermint.**

| S.O.V | df | fo | YII | Fv/fm | fm | fv | Y(NPQ) | Y(NO) | NPQ | ETR |
|---|---|---|---|---|---|---|---|---|---|---|
| Replication | 2 | 0.0017 | 0.00337 | 0.00116 | 0.7396 | 0.7337 | 0.0004 | 0.0011 | 0.0045 | 14.7708 |
| Salinity (S) | 3 | 0.6930** | 0.14945** | 0.09022** | 13.4999** | 20.2384** | 0.0446** | 0.0456** | 0.3145** | 300.055** |
| Fungi (F) | 3 | 0.0412** | 0.01461** | 0.00362** | 0.4269** | 0.6231** | 0.0054** | 0.0031** | 0.0153** | 41.50** |
| S × F Interaction | 9 | 0.0047** | 0.00097** | 0.00049** | 0.1005$^{ns}$ | 0.0988$^{ns}$ | 0.0015$^{ns}$ | 0.0005$^{ns}$ | 0.0042$^{ns}$ | 3.7037$^{ns}$ |
| Error | 30 | 0.0015 | 0.00034 | 0.00013 | 0.0772 | 0.0736 | 0.00072 | 0.00026 | 0.0027 | 3.304 |
| CV% | | 3.04 | 3.26 | 1.52 | 4.83 | 6.11 | 15.44 | 8.3 | 5.42 | 3.304 |

ns = no significant

* significant (P < 0.05)

** significant (P < 0.01)

but also a mycelial network of AMF and *P. indica* was spread around the roots (Fig 2). The investigation into the AMF and *P. indica* and co-inoculation of them with plants showed 87%, 77%, and 89% colonization rate, respectively. Salt stress significantly reduced the ability of endophytic fungi to colonize with peppermint root. The decline was more noticeable when plants were inoculated with *P. indica* under medium level of salinity (6 dS m$^{-1)}$. Generally, the highest decline in the percentage of root colonization in both fungi (AMF and *P. indica*) was observed in the 9 dS m$^{-1}$ level of salinity. The synergistic association of *P. indica* and AMF considerably increased the colonization ability of them with peppermint roots even under intense salt stress (Fig 3).

## 3.2. Drought stress increases membrane Electrolyte Leakage (EL)

Membrane electrolyte leakage was increased under salinity stress by up to 70.4% (9 dSm$^{-1}$) (Fig 4 and S1 Table), while microbial inoculation significantly alleviated oxidative damage to the membrane permeability even under severe salt stress. However, the difference between *P. indica* and AMF was not significant. The decrease in the percentage of EL was more noticeable when plants were co-inoculated by both *P. indica* and AMF. The results showed that co-inoculation was more effective than either single inoculation with *P. indica* or AMF for reducing membrane electrolyte leakage in the high level of salinity (9 dSm$^{-1}$).

**Table 3. Variance analysis of salinity and *P. indica* and AMF effects on measured physiological and biochemical parameters of peppermint.**

| S.O.V | df | Chlorophyll a (µg/ml) | Chlorophyll b (µg/ml) | P (meq/gdw$^{-1}$) | K$^+$ (meq/gdw$^{-1}$) | Na$^+$ (meq/gdw$^{-1}$) | Essential oil (%) | Stomatal conductance (mmol (H$_2$O) m$^{-2}$ s$^{-1}$) | Membrane electrolyte leakage % | colonization rate (%) |
|---|---|---|---|---|---|---|---|---|---|---|
| Replication | 2 | 0.3765 | 0.50701 | 0.00255 | 0.02985 | 0.0043 | 0.00316 | 0.0593 | 2.9203** | 11.583 |
| Salinity (S) | 3 | 13.9547** | 5.57805** | 0.12515** | 3.99761** | 2.1872** | 6.81303** | 655.0106** | 4835.1701** | 2267.629** |
| Fungi (F) | 3 | 1.7688** | 1.00595** | 0.07111** | 0.15185** | 0.0690** | 0.82588** | 28.2901** | 133.2135** | 424.750** |
| S × F Interaction | 9 | 0.0399$^{ns}$ | 0.07208$^{ns}$ | 0.00281$^{ns}$ | 0.02509$^{ns}$ | 0.0145* | 0.04237** | 0.4364* | 13.5865** | 13.824* |
| Error | 30 | 0.08101 | 0.1196 | 0.00227 | 0.02054 | 0.0061 | 0.0068 | 0.1701 | 6.0838 | 5.2500 |
| CV% | | 5.03 | 6.95 | 14.21 | 13.2 | 14.12 | 4.59 | 3.34 | 6.44 | 3.25 |

ns = no significant

* significant (P < 0.05)

** significant (P < 0.01)

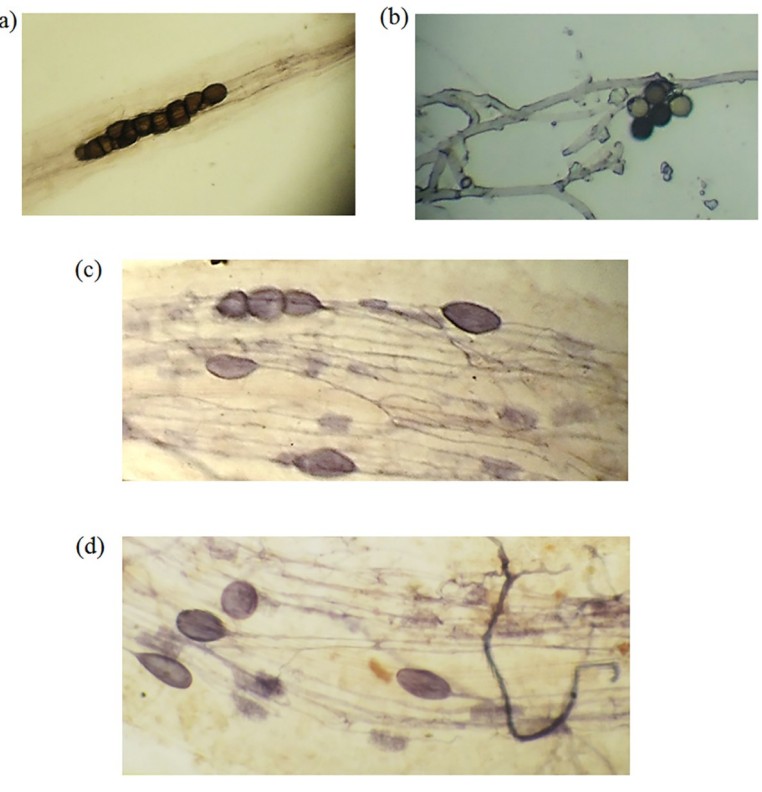

**Fig 2.** *Piriformospora indica* interacellular chlamydospores in cortex (a), *P. indica* spore and hyphae (around the roots) (b) and interadical hyphae, exteradical hyphae, vesicles and arbuscules formation of AMF (c and d) were formed in the *Mentha piperita* roots. The root-pieces were examined under light microscope at the magnification of 10–40 X.

## 3.3. Chlorophyll fluorescence parameters

### 3.3.1. Primary fluorescence (Fo), maximum fluorescence (Fm), variable fluorescence (Fv).
It is known that salinity stress causes photosynthetic apparatus injury. Our data showed that salinity and microbial inoculation remarkably affected the Chlorophyll fluorescence; as salt stress led to a remarkable increase in Fo value, with maximum enhancement (29.6% and 28.9%, respectively compared to the control treatment) in plants which were grown under medium and high levels of salinity (Fig 5 and S1 Table). In addition, the increase in Fo was consistently accompanied by a decline in Fm and Fv values in salinity-treated leaves compared

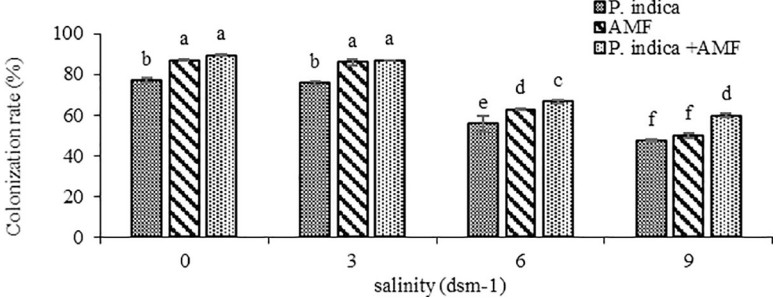

**Fig 3. Effect of salinity on root colonization (*P. indica,* AMF and co-inoculation) in peppermint.** In each figure, means with the same letter are significantly different according to LSD test at P < 0.05.

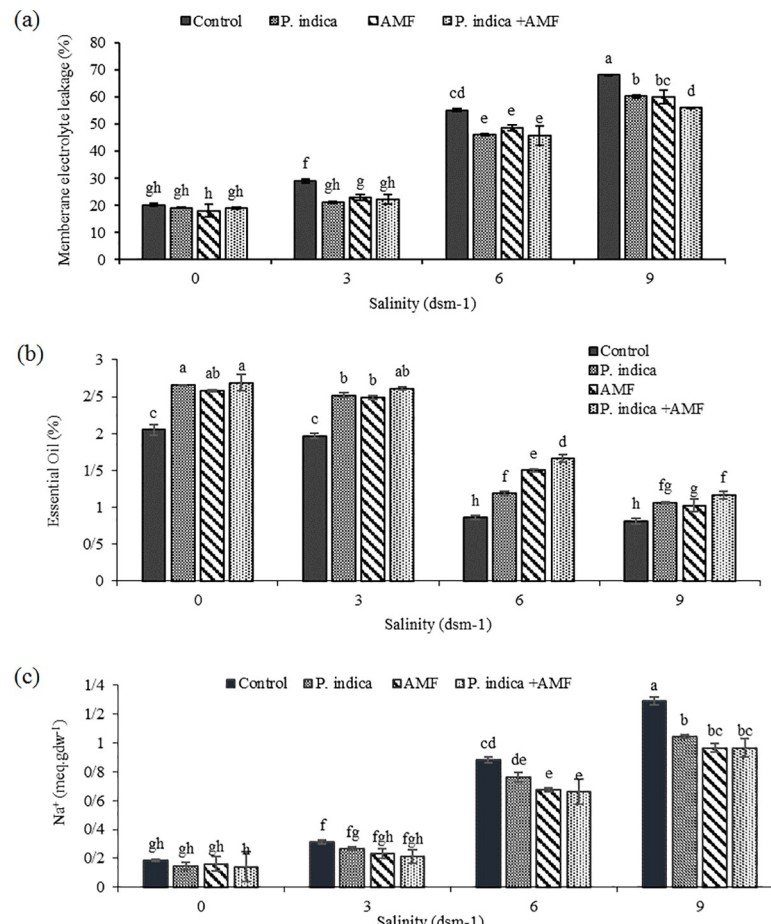

**Fig 4.** Effect of salinity on Membrane electrolyte leakage (a), Essence (b) and Na (c) in inoculated and non-inoculated peppermint with (*P. indica* AMF and co-inoculation).

to the control (33.3%, and 48.81% reduction, respectively, when plants were subjected to the severe stress) (Fig 6 and S2 Table). As indicated in the Fig 5, symbiotic relationship with *P. indica* and AMF and their co-inoculation significantly reduced Fo, while the same treatments significantly enhanced Fm and Fv. The highest Fm and Fv were observed in *P. indica* -inoculated plants (7.09%, and 24.33% higher, respectively compared to the control) (Fig 6 and S3 Table).

**3.3.2. Maximum photochemical quantum yield of PSII (Fv/Fm) was reduced under salt stress.** The Fv/Fm ratio considerably decreased when plants were subjected to salinity stress; the reduction was even greater when the higher salinity level (9 dSm$^{-1}$) was applied (Fig 5). As expected, the symbiotic relationship with endophytic fungi was associated with the increased Fv/Fm ratio under both normal and salinity conditions. This improvement was more noticeable when salinity reached 9 dS m$^{-1}$, as almost increased by 11.47%, 9.83%, and 6.55%, with single and co-inoculation with *P. indic*a and AMF respectively compared to the control plants (Fig 5 and S1 Table).

**3.3.3. Salt stress significantly reduced photochemical efficiency of photosystem II (ΦPSII) and ETR.** In this experiment, ΦPSII and ETR values were sharply decreased under saline condition (S2 Table), with maximum reduction (39.7% and 53.36% lower respectively, compared to the control) in plants grown under high salinity (9 dSm$^{-1}$) (Figs 5 and 6). In

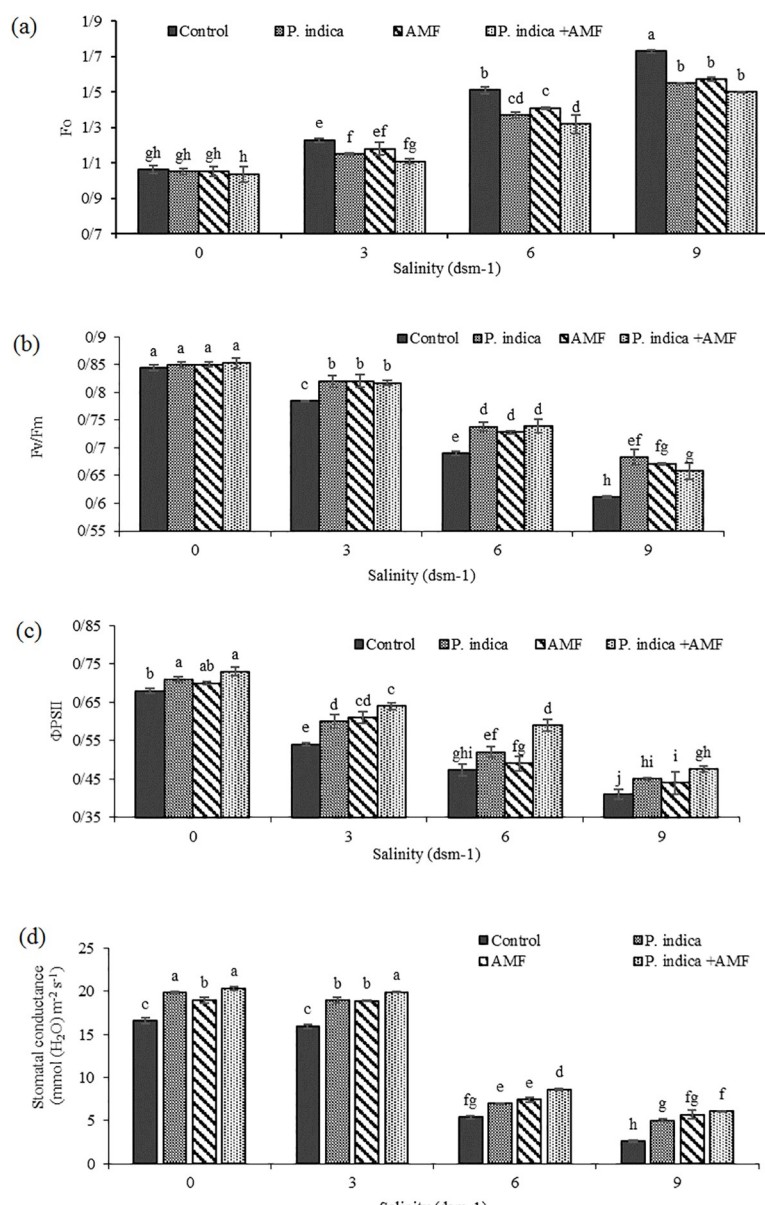

**Fig 5.** Effect of salinity on Fo (a), ΦPSII (b), Fv/Fm (c) and Stomatal conductance (d) in inoculated and non-inoculated peppermint with (*P. indica*, AMF and co-inoculation).

general, endophytic fungi and especially their co-inoculation significantly improved the ability of plants to respond to the negative effects of salinity, with the most obvious improvement of ΦPSII in 9 dSm$^{-1}$, (13.86% compared to control plants) (Fig 5). ETR was also positively affected by endophytic fungi inoculation, as the highest ETR (24.11%) was observed in co-inoculated plants (Fig 6 and S3 Table)

 **3.3.4. Nonphotochemical chlorophyll quenching (NPQ), YNPQ and YNO were increased under salt stress.** Salinity had significant effects on NPQ, YNPQ and YNO values in peppermint plants. The plants subjected to salt stress showed a remarkable increase in NPQ, YNPQ and YNO values in comparison to the control plants. The highest NPQ, YNPQ and

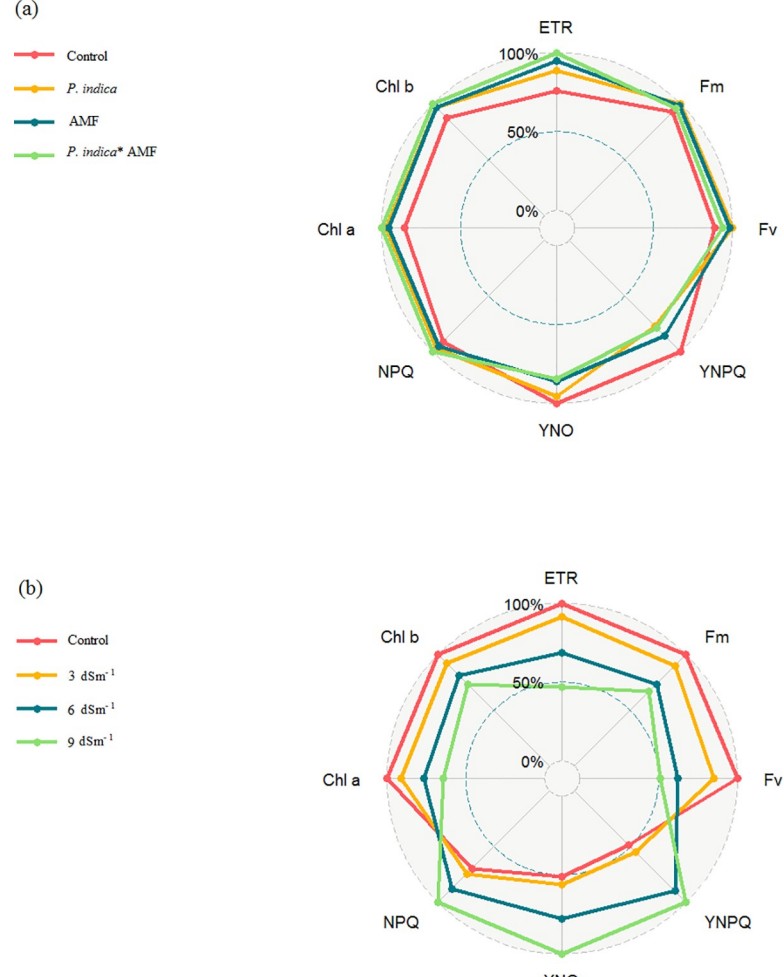

**Fig 6.** Changes in various physiological traits in inoculated and non-inoculated peppermint with (*P. indica*, AMF and co-inoculation) (a), and caused by salinity stress in peppermint plant (b) *P. indica*; AMF–Mycorrhizae treatment; AMF* *P. indica*–Mycorrhizae and *P. indica* treatment.

YNO values (30.76%, 52.17% and 49.09%, respectively compared to controlled plants) was observed in plants which were grown under salt stress (9dsm$^{-1}$) (Fig 6 and S2 Table).

The results indicated that the fungi inoculation significantly reduced the NPQ, YNPQ and YNO values compared to controlled plants. The most reduction was observed in co-inoculated plants (Fig 6 and S3 Table).

**3.3.5. Stomatal conductance (gs) is decreased under salt stress.** The effect of saline condition on the stomatal conductance (gs) in the peppermint plants is shown in Fig 5. Increasing salinity led to a reduction in stomatal conductance. As indicated in Fig 5, gs reached to the lowest level under 9 dSm$^{-1}$ of salinity (84.2% lower compared to the control). In contrast, fungi inoculation remarkably increased gs (53.87%, 47.91%, and 57% higher in *P. indica*, AMF and co-inoculation, respectively compared to un-inoculated plants grown under salt stress (9dsm$^{-1}$)) (Fig 5 and S1 Table).

**3.3.6 Salt stress reduces chlorophyll content.** Salt stress led to a remarkable decrease in chlorophyll a and b compared to the control. The lowest contents of chlorophyll a and b, were observed in the plants treated with sever salinity (35.82% and 26.87%, respectively compared

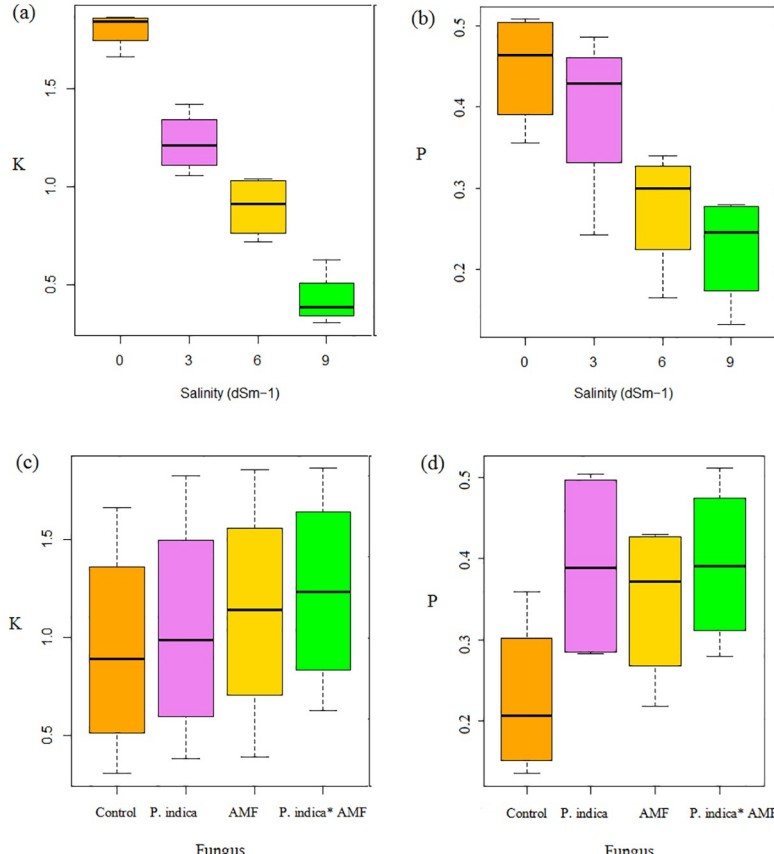

**Fig 7.** Box Plots of P and K in peppermint under salinity stress (a) and in inoculated and non-inoculated peppermint with (*P. indica*, AMF and co-inoculation) (b). *P. indica*; AMF–Mycorrhizae treatment; AMF* *P. indica*–Mycorrhizae and *P. indica* treatment.

to the control plants) (Fig 6 and S2 Table). As it can be seen in Fig 6 and S3 Table, fungi inoculation and especially co-inoculation significantly enhanced chlorophyll a and b contents (14.71% and 12.66%, respectively compared to controlled plants).

**3.3.7. P, K, Na.** The effect of saline condition on the minerals (P, $K^+$ and $Na^+$) concentrations in the peppermint plants is shown in Figs 4 and 6. According to the results, Na content was dramatically increased in the peppermint leaves when plants were exposed to salt stress. Fungi inoculation ameliorated salinity negative effects, and the alleviation ratio (25.11% lower than control plant) was more considerable in co-inoculated plants under high (9 $dSm^{-1}$) level of salinity (Fig 4 and S1 Table). However, the opposite trend was observed for $K^+$ and P, as increasing in Na level was accompanied by a decline in P and $K^+$ content. This reduction was even greater when the higher salinity concentration (9 $dSm^{-1}$) was used (49.53% and 76.27% respectively, compared to the control) (Fig 7 and S2 Table). Our data shows that microbial inoculation positively affected the absorption of P and K in rhizosphere soil. The highest P content (43.58%) was observed in co-inoculated plants (Fig 7 and S3 Table).

**3.3.8. Principal component of chlorophyll fluorescence parameters.** Principal component analysis was conducted on the data of the salinity stress × 9 traits selected of leaf chlorophyll fluorescence parameters. The association between the traits were compared across the control, AMF, *P. indica* and co-inoculation treatments (Fig 8). The score plot displayed four different groups, which were associated with four salinity samples, indicating a distinct

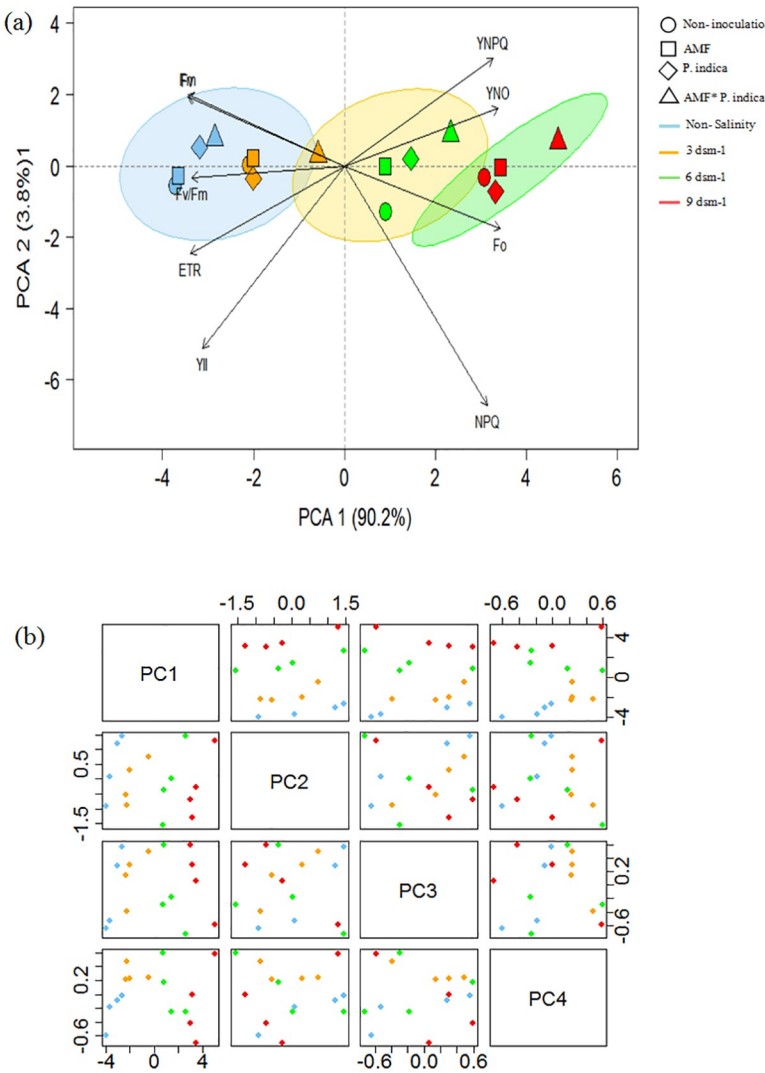

**Fig 8.** Principal component analysis (PCA) (a) and 2D score plot (b) of chlorophyll fluorescence parameters in peppermint leaves. The score plot for the four-salinity levels; 0, 3, 6 and 9 dsm$^{-1}$ were indicated in blue, orange, green and red, respectively. Abbreviations: *P. indica*; AMF–Mycorrhizae treatment; *P. indica*\*AMF–*P. indica* and Mycorrhizae treatment.

separation of the chlorophyll fluorescence profiles under salinity conditions (Fig 8). The PC1 mainly explained the separation of samples caused by 6 and 9 sm$^{-1}$ of salinity (inoculation and non-inoculation), which was significantly positively correlated with YNPQ, YNO, Fo and, NPQ; whereas, PC2 mainly described non-stressed peppermint plants under low and mild salinity (inoculation and non-inoculation), it also showed the positive correlation with the Fm, Fv, Fv/Fm, ETR and, ΦPSII. Additionally, PCA findings, showed a tight positive association between Fm and Fv while they were associated negatively with Fo.

**3.3.9. Essential oil content and composition.** Salinity stress negatively affected essential oil content. Generally, salinity levels decreased essential oil content, and the lowest amount was recorded under 9 dSm$^{-1}$. Surprisingly, fungal symbiosis had a positive impact on essential oil content under all evaluated conditions. However, when the higher salinity concentration (9 dSm$^{-1}$) was used, co-inoculation was more effective in improving essential oil content compared to those of single inoculated plants (Fig 4 and S1 Table).

**Table 4. Effect of microbial inoculation on *Mentha piperita* essential oils composition under salinity stress.**

| No | Compound | Concentration (%) | | | | | | | |
|----|----------|-------------------|---|---|---|---|---|---|---|
| | | Salinity* *P. indica* *AMF | Salinity*AMF | Salinity* *P. indica* | Salinity | *P. indica* *AMF | *P. indica* | AMF | control |
| 1 | limonene | 2.4 | 1.93 | 1.42 | 3.5 | 2.63 | 2.89 | 2.73 | 5.36 |
| 2 | Carvacrol, methyl ether | 1.38 | 1.77 | 2.08 | 1.23 | 2.01 | 2.21 | 1.9 | 1.38 |
| 3 | Menthyl acetate | 1.21 | 1.13 | 1.22 | 1.07 | 1.12 | 1.16 | 1.15 | 1.08 |
| 4 | α-Pinene | 1.04 | 1.07 | 1.09 | 1.03 | 1.05 | 0.93 | 1.02 | 0.85 |
| 5 | sabinene | 0.92 | 0.88 | 0.75 | 0.72 | 0.82 | 0.84 | 0.82 | 0.86 |
| 6 | β—Pinene | 0.92 | 0.51 | 0.61 | 0.43 | 0.61 | 0.6 | 0.63 | 0.52 |
| 7 | β—Bourbonene | 0.72 | 0.65 | 0.78 | 0.41 | 0.43 | 0.38 | 0.54 | 0.32 |
| 8 | Terpinolene | 0.43 | 0.14 | 0.35 | 0.31 | 0.49 | 0.48 | 0.53 | 0.81 |
| 9 | α- Terpinene | 0.28 | 0.27 | 0.23 | 0.23 | 0.41 | 0.48 | 0.42 | 0.64 |
| 10 | cis-Ocimene | 0.14 | 0.29 | 0.6 | 0.33 | 0.52 | 0.62 | 0.54 | 0.64 |
| 11 | Piperitone | 0.1 | 0.31 | 0.13 | 0.29 | 0.23 | 0.13 | 0.21 | 0.28 |
| 12 | Menthol | 19.87 | 19.37 | 18.03 | 15.39 | 18.12 | 18.89 | 17.3 | 14.86 |
| 13 | neoisomenthol | 5.41 | 5.27 | 6.18 | 4.46 | 5.37 | 6.09 | 5.35 | 5.07 |
| 14 | Trans sabinene hydrate | 0.92 | 0.61 | 0.98 | 0.68 | 0.87 | 0.81 | 0.8 | 0.85 |
| 15 | 1,8-Cineole | 10.66 | 9.62 | 9.23 | 9.01 | 9.14 | 9.04 | 9.12 | 8.87 |
| 16 | Menthofuran | 5.22 | 6.18 | 5.07 | 5.87 | 5.63 | 5.61 | 5.48 | 7.33 |
| 17 | α-Terpineol | 0.87 | 0.52 | 0.85 | 0.58 | 0.75 | 0.63 | 0.72 | 0.61 |
| 18 | Linalool | 0.68 | 0.54 | 0.26 | 0.56 | 0.5 | 0.49 | 0.47 | 0.52 |
| 19 | Menthone | 34.52 | 31.27 | 32.98 | 31.04 | 34.75 | 34.95 | 35.2 | 29.41 |
| 20 | iso-menthone | 4.9 | 5.74 | 4.57 | 6.15 | 6.12 | 5.95 | 5.97 | 8.34 |
| 21 | Pulegone | 0.23 | 0.33 | 0.34 | 0.24 | 0.27 | 0.3 | 0.26 | 0.15 |
| 22 | Caryophyllene | 0.81 | 0.36 | 0.76 | 0.45 | 0.75 | 0.73 | 0.75 | 0.68 |
| 23 | β—Farnesene | 0.37 | 0.33 | 0.35 | 0.32 | 0.25 | 0.28 | 0.27 | 0.31 |
| 24 | germacrene D | 0.17 | 0.21 | 0.21 | 0.17 | 0.2 | 0.25 | 0.22 | 0.15 |
| 25 | Caryophyllene oxid | 0.19 | 0.13 | 0.14 | 0.11 | 0.11 | 0.12 | 0.1 | 0.06 |
| 26 | α-Humulene | 0.15 | 0.11 | 0.14 | 0.18 | 0.15 | 0.27 | 0.1 | 0.16 |
| 27 | Viridiflorol | 0.15 | 0.13 | 0.12 | 0.11 | 0.18 | 0.31 | 0.21 | 0.43 |

Salinity (9dsm$^{-1}$); *P. indica*; AMF–Mycorrhizae treatment; *P. indica*

*AMF–*P. indica* and Mycorrhizae treatment.

Essential oil composition was affected by *P. indica*, AMF, co-inoculation, salinity, and their interactions (Table 4 and Fig 10). In all essential oil samples, the major constituents were Menthone, Menthol, 1,8-Cineole, and iso-menthone, respectively. As it can be seen in Table 4, the pattern of Menthone and Menthol increase differed in stress and non-stress conditions, indicating that microbial inoculation positively enhanced Menthone and Menthol content in both conditions. The highest Menthone content was recorded in the AMF-inoculated plant's essential oil, whereas the highest Menthol content as the second main component was observed in co-inoculated plants grown under salt stress (Table 4). A similar pattern was also observed for 1,8-Cineole, while the pattern for Menthofuran, another major component, was completely different, where salinity and Microbial inoculation reduced this content compared to the control plants (Table 4).

Piperitone content was increased under high salinity stress (9 dSm$^{-1}$) while β–Pinene and Terpinolene decreased compared with control samples under the same salinity level. Under both control and salinity conditions, fungi inoculation enhanced α-Pinene, β–Pinene, 1,8-Cineole, Menthone, Menthol, neoisomenthol, Pulegone, α-Terpineol, Menthyl acetate, Carvacrol,

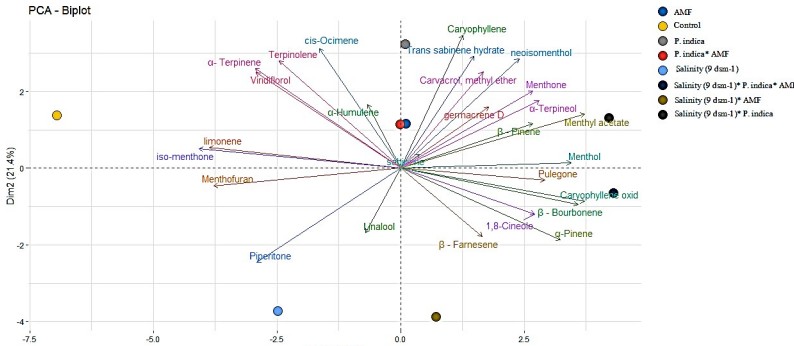

**Fig 9. Principal Component Analysis (PCA) of metabolites in peppermint leaves.** AMF–Mycorrhizae treatment.

methyl ether, β–Bourbonene, Caryophyllene, and germacrene D. In addition, higher levels of Menthone, Menthol and 1,8-Cineole were considerable in co-inoculated plant under salinity stress.

**3.3.10. Heatmap and principal component of the essential oil constituents.** As it can be seen in Figs 9 and 10, Dim1 or PC1 accounted for 42.3% of the total variation and was mainly influenced by hydrocarbon monoterpenes (which is mainly due to the relative percentage variation of menthol), hydrocarbon sesquiterpenes, and alcohol monoterpenes. The PC1 mainly explained the separation of samples caused by endophytic fungi (salinity and non-salinity conditions), whereas, PC2 mainly explained salinity-treated and control peppermint plants (Fig 9).

A Heatmap analysis was carried out in order to identify the significantly changed metabolites between inoculated and non-inoculated peppermint under salinity and non-salinity conditions (Fig 11). Interestingly, salinity and non-salinity treatment (inoculated plants and non-inoculated plants) were separated from each other by two opposing groups (Fig 11). Based on cluster analysis, two clusters of metabolites can be identified; all metabolites of the PC1, except α-Humulene, were classified in the first cluster, whereas the other group consisted of the PC2.

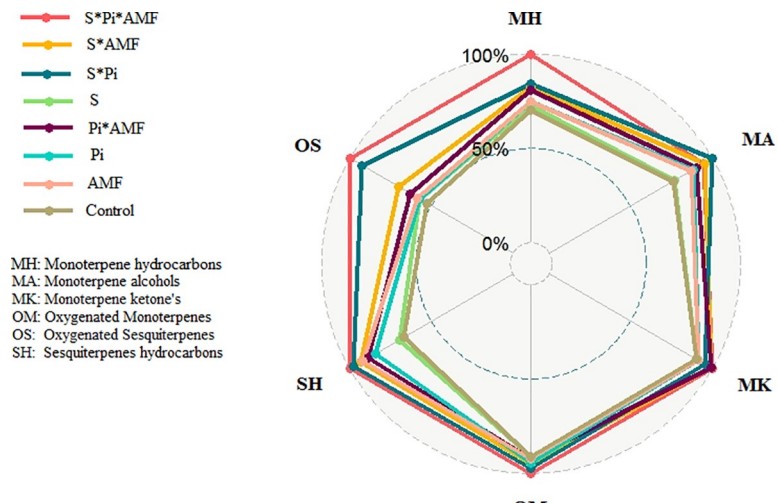

**Fig 10. Changes in terpenoid content in inoculated and non-inoculated peppermint with (*P. indica*, AMF and co-inoculation) under salinity stress in peppermint plant.** S–Salinity; *P. indica*; AMF–Mycorrhizae treatment; AMF* *P. indica*–Mycorrhizae and *P. indica* treatment.

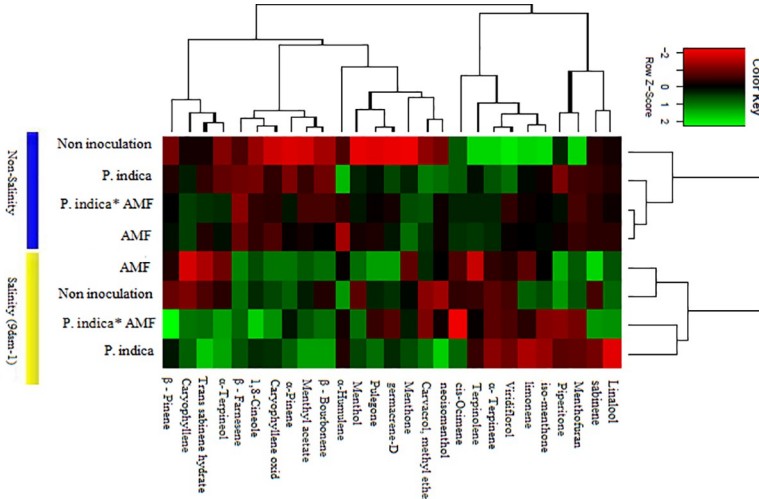

**Fig 11. Heatmap pattern of the essential oil constituents from peppermint.** AMF–Mycorrhizae treatment; *P. indica* *AMF–*P. indica* and Mycorrhizae treatment.

However, linalool, Piperitone and Menthofuran were not included in either of the first two components. Controlled plants contained the higher iso-menthone, Terpinolene, and limonene together with Menthofuran and Viridiflorol which is related to oxygenated terpenes. It is possible to observe that co-inoculation treatment largely altered the relative percentage proportion of the essential oil compounds, with a high relative percentage of Menthol, Menthone, β—Pinene, 1,8-Cineole, Caryophyllene oxid, α-Terpineol, Menthyl acetate, β—Bourbonene, linalool, β–Farnesene, and sabinene. However, some metabolites showed higher percentages in control plants and co-inoculation had no significant effect on their production. The significantly changed metabolites in peppermint leaves under salinity stress were shown in Fig 12.

## 4. Discussion

The reduction of root colonization by *P. indica* and AMF in response to increasing salinity (6 to 9 dSm−1) observed in the current study is similar to previous reports in other plants [35, 36]. It is well documented that oxidative damage to the thylakoid membrane under salinity induces generation of ROS and impairs electron transport function [37]. In the current study, salt stress decreased ETR, Fm, Fv/Fm ratio, and ΦPSII, while the Fo and NPQ rates increased under salinity conditions. Such alterations in fluorescence parameters can mainly be linked to the impairment of PSII function and inhibition of the electron transfer process. In our study, it may probably be due to 1: the membrane system damage in peppermint, which increases membrane electrolytic leakage (Fig 4), and 2: an increase in toxic ions such as Na+ (Fig 4).

 A decrease of Fm in response to salinity was along with a reduction in chlorophyll content (Fig 6). The Fm reduction may be linked to the deactivation of chlorophyll protein compounds due to leaf chlorosis as well as alterations in biochemical reactions [38]. Similar findings of increased Fo and decreased Fm are also reported in maize under salt stress [39], and low-temperature stress [40], and in winter wheat under drought stress [8]. An increase in Fo rate can indicate potential damages to the PS II; damages include D1 protein degradation, a decrease in thylakoid membrane integrity, damages to the photosystem II electron transport chain, the reduction of plastoquinone electron receptors, the lack of complete oxidation of plastoquinone, and the separation of light-harvesting protein complexes in chlorophyll a/b [41, 42]. The current observation of a reduction in Fv/Fm under salinity stress can be due to the limitation

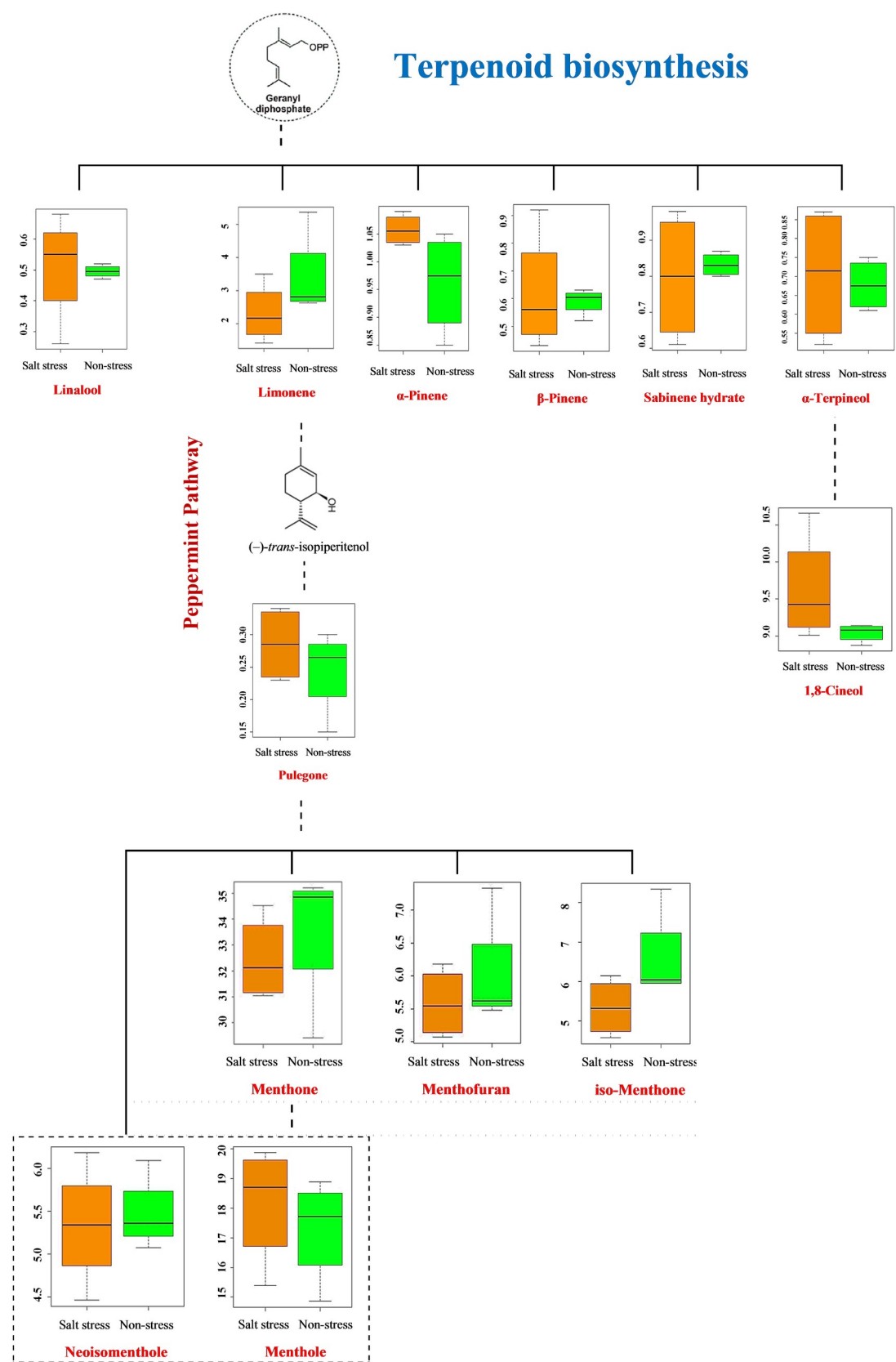

**Fig 12. Leaf metabolites involved in the terpenoied pathways in peppermint plant under salinity stress.** The relative abundance of metabolites in inoculated and non-inoculated peppermint with (*P. indica* and AMF) under salt stress was shown as box plots. Regulated metabolites under salinity and non-salinity condition were indicated in orange and green, respectively.

in the electron transport chain, which can destruct PSII reaction centers and reduce PSII maximum efficiency [9, 43]. In the present experiment, the decrease in chlorophyll was accompanied by a decline in $\Phi$PSII (Fig 11), which can be explained by a lower energy transfer efficiency from antenna chlorophylls to the photosystem II reaction centers.

Similar findings were reported in other plants in which salinity induced changes [44, 45]. It is known that reduction in the chlorophyll content under salt stress reduces light reactions and electron transport in PS II from primary to secondary acceptor [46]. Furthermore, there was an inverse relationship between $\Phi$PSII and NPQ value in peppermint ($r^2 = 0.81$; $p < 0.001$) (Fig 13). The decrease in $\Phi$PSII and the significant increase in NPQ in peppermint leaves under salinity stress indicates a decrease in photosynthetic process and carbon fixation capacity. Such impairment in photosynthetic process can lower utilization of electron transport products which leads to a greater thermal dissipation of light energy [47]. It is assumed that although salinity affected the rate of colonization of *P. indica* and AMF and their co-inoculation symbiosis in peppermint, the presence of these fungi in the plant's roots improves the function of the plant photosynthetic apparatus. It can be due to the fact that symbiotic fungi improve PSII function in plants under salinity stress [48]. Similar positive effects of plant-microbe interactions on PSII photochemical activity were recorded in many other plants [49, 50]. The positive effect of endophytic fungi against salinity impacts on chloroplast and chlorophyll in peppermint can also be due to fungal stimulation of antioxidant synthesis in the plant or due to an improvement in ROS scavenging activity of the newly synthesised secondary metabolites such as phenolic compounds or alleviation of damage to the cell membranes caused by salt stress [18, 51, 52]. The observed correlation between increasing chlorophyll content and high $\Phi$PSII in microbial treated plants under salinity stress reveals that the fungi

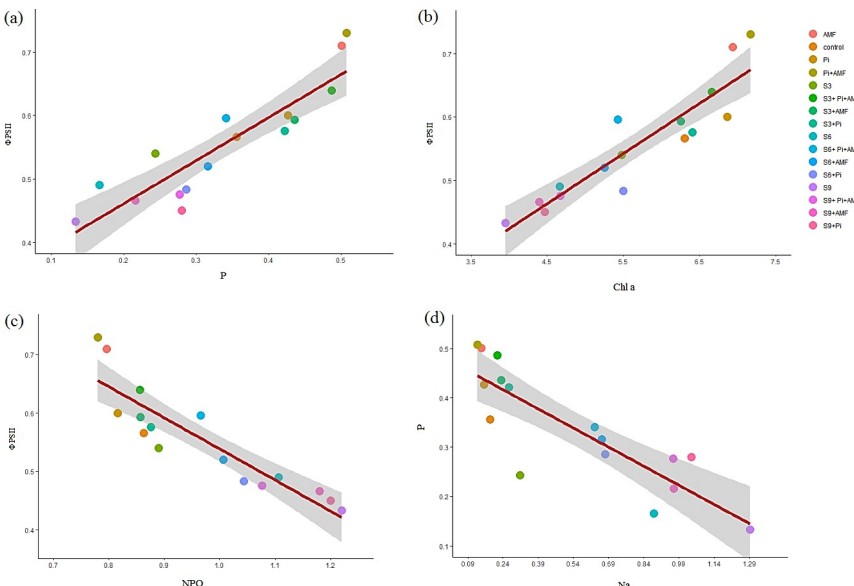

**Fig 13. Relationship between some physiological parameters in inoculated and non-inoculated peppermint with (*P. indica*, AMF and co-inoculation) under salinity stress in peppermint plant.** AMF–Mycorrhizae treatment; AMF* *P. indica*–Mycorrhizae and *P. indica* treatment.

alleviate the adverse effects of salinity on light-harvesting and electron transfer to plastoquinone through stimulation of chlorophyll accumulation in the plants.

A remarkable reduction in P and K+ uptake observed in non-inoculated plants under severe salinity stress might be because of the high rate of Na+ absorption. Moreover, the inhibitory effect of ion toxicity, such as Na+ and Cl−, on the mineral absorption directly influences photosynthetic performance. Such a direct correlation of nutrient imbalance and a simultaneous decline in PSII function in salt-stressed plants is already reported [53, 54]. The reason could be a reduction in membrane permeability leading to an increasing rate of ion leakage (Fig 4), and antagonistic relationship between excess uptake of Na+ and mineral nutrient such as P and K + under severe salinity stress (Fig 13). The same can lead to an imbalance in cellular ion homeostasis [37]. The results show that the endophytic fungal symbiosis increased P and K+ content (Fig 6), but decreased Na+ uptake in peppermint leaves (Fig 4). The observation of a positive relationship between the chlorophyll and gas exchange with PSII efficiency (Fig 13) can mainly be through elevation in the mineral absorption and improvement in water status [55]. It is also well documented that AMF inoculation positively influences chlorophyll biosynthesis by alleviating the detrimental effects of salinity on $Mg^{2+}$ absorption and improves Fv/Fm values [56].

In the present experiment, a correlation observed between the ETR and some essential oil (Fig 14) confirms the fact that monoterpene emissions maintain PSII stability in stressed plants

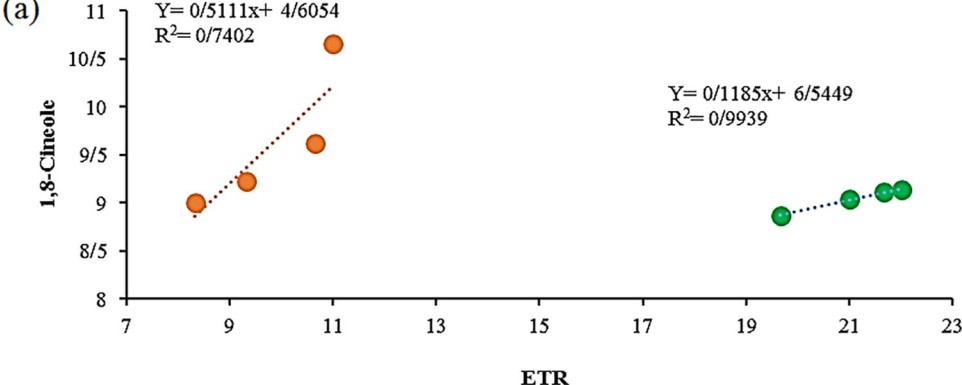

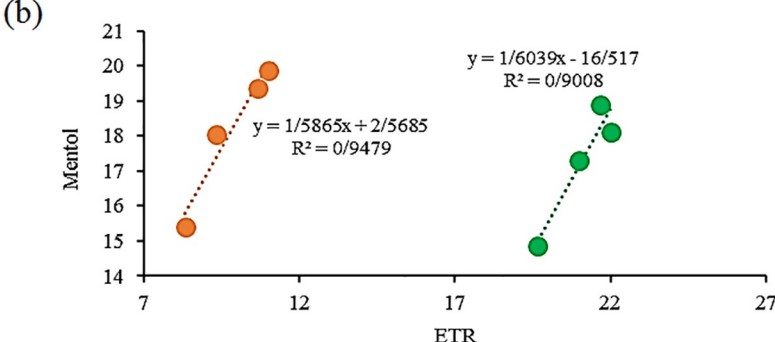

**Fig 14. Relationship between some metabolites in inoculated and non-inoculated peppermint with (*P. indica*, AMF and co-inoculation) under 9 dsm⁻¹ salinity stress in peppermint plant.** The salinity and non-salinity condition were indicated in orange and green, respectively.

by partitioning excess electrons into other sinks of the terpenoid biosynthetic pathway [57]. A similar relationship between monoterpene emission, increasing Fv/Fm ratio, and ETR was reported in many other plants under different kinds of stress [57, 58]. Besides, high ETR values observed in mycorrhizal plants can be attributed to a better physiological status, changes in the pattern of carbon partitioning, and accumulation and biosynthesis of terpenes. The enhanced biosynthesis of terpenoids in endophytic fungi treated plants can be attributed to the role of the fungi in mineral nutrients uptake [22, 59].

## 5. Conclusion

Co-inoculation of mycorrhiza and *Piriformospora indica* fungi had a synergic effect on the enhancement of peppermint essential oil and its physiological characteristics. Fungal co-inoculation protected peppermint plants under salinity conditions. There are many reports about the synergistic effect of symbiotic fungi on the improvement of plant function under various environmental conditions; however, there are other research which did not support the synergistic effect and emphasized that separate application of endophyte fungi had more significant effect on the antioxidant activity and Fv / fm [60, 61]. Thus, it needs further research in the future because applying microbial treatments for boosting plant secondary metabolite production could be a natural way as well as a sustainable approach in the promising herbal medicine industry.

## Supporting information

**S1 Table. Effect of salinity on membrane electrolyte leakage, stomatal conductance, essential oil, Na+, Fv/Fm, YII and Fo in inoculated and non-inoculated peppermint with (*P. indica* AMF and co-inoculation).**
(DOCX)

**S2 Table. Effect of salinity on fm, fv, chlorophyll a, chlorophyll b, Y(NPQ), Y(NO), P, K+, NPQ and ETR.**
(DOCX)

**S3 Table. Changes in various physiological traits in inoculated and non-inoculated peppermint with (*P. indica*, AMF and co-inoculation).**
(DOCX)

**S1 Graphical abstracts.**
(TIF)

## Author Contributions

**Data curation:** Masoumeh Khalvandi.

**Formal analysis:** Masoumeh Khalvandi, Mohammadreza Amerian, Hematollah Pirdashti, Sara Keramati.

**Investigation:** Masoumeh Khalvandi, Mohammadreza Amerian, Hematollah Pirdashti, Sara Keramati.

**Methodology:** Masoumeh Khalvandi, Mohammadreza Amerian, Hematollah Pirdashti.

**Project administration:** Hematollah Pirdashti.

**Resources:** Mohammadreza Amerian.

**Software:** Masoumeh Khalvandi, Sara Keramati.

**Supervision:** Mohammadreza Amerian, Hematollah Pirdashti.

**Writing – original draft:** Masoumeh Khalvandi, Sara Keramati.

**Writing – review & editing:** Masoumeh Khalvandi, Mohammadreza Amerian, Hematollah Pirdashti, Sara Keramati.

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
