## [Decision Letter · Decision Letter 0]

24 Feb 2021

PONE-D-21-02576

Does co-inoculation of mycorrhiza and Piriformospora indica fungi enhance the efficiency of Chlorophyll fluorescence and essential oil composition in peppermint under irrigation with saline water from the Caspian Sea?

PLOS ONE

Dear Dr. Khalvandi,

Thank you for submitting your manuscript to PLOS ONE. After careful consideration, we feel that it has merit but does not fully meet PLOS ONE’s publication criteria as it currently stands. Therefore, we invite you to submit a revised version of the manuscript that addresses the points raised during the review process.

We look forward to receiving your revised manuscript.

Kind regards,

Mayank Gururani

Academic Editor

PLOS ONE

Journal Requirements:

5. Please ensure that you refer to Figures 7, 10 and 12 in your text as, if accepted, production will need this reference to link the reader to the figures.

Reviewers' comments:

Reviewer's Responses to Questions

**Comments to the Author**

1. Is the manuscript technically sound, and do the data support the conclusions?

Reviewer #1: Yes

Reviewer #2: Yes

Reviewer #3: Yes

2. Has the statistical analysis been performed appropriately and rigorously? 

Reviewer #1: Yes

Reviewer #2: Yes

Reviewer #3: Yes

3. Have the authors made all data underlying the findings in their manuscript fully available?

Reviewer #1: Yes

Reviewer #2: No

Reviewer #3: Yes

4. Is the manuscript presented in an intelligible fashion and written in standard English?

Reviewer #1: Yes

Reviewer #2: Yes

Reviewer #3: Yes

5. Review Comments to the Author

Reviewer #1: Critical comments

General comments

This is a well written article explaining the results of a well conducted experiment on synergistic influence of AM and Pirimorphospora sp. on salinity stress in Peppermint. The article has impressive findings, acceptable for publication in the esteemed Plos-1 journal. However, the language need minor corrections. It seems that the authors have some self-contradictory arguments in the discussion. The authors may do minor corrections giving due credit to the specific comments given below.

Specific comments

Page 1: Introduction: line 1-2: “which its demand has grown dramatically in recent decades” – may be corrected as: ‘the demand of which has grown dramatically in recent decades’

Line 3: “It is evident that peppermint essential oil” – may be modified as: ‘it is evident that the essential oil’

Line 5-6: “as well as the reactive oxygen species (ROS) scavenging activity because of its valuable constituents” – ‘as well as scavenging of reactive oxygen species (ROS)’

2nd Para – line 1: - avoid ‘Nowadays’; the sentence may begin as ‘The sustainability …’

Line 5: no need of repeating – ‘reactive oxygen species (ROS)’, better put the short form only – ROS because the same is already defined above

Line 7: “stresses; it can show’ – ‘stresses, because the same can show’

Page 3: 2nd para – 2nd line: not ‘sever’ – but ‘severe’

Page 4: para 3- line 2: not ‘fungi symbiosis’ – ‘fungal symbiosis’

Discussion:

Page 4-5: line 1-4: “In the present study, the root colonization with P. indica and AMF were remarkably reduced by increasing salinity (6 to 9 dSm−1). Similar observations were reported in other plants (Wu et al., 2010; Hadian-Deljou, et al, 2020; khalvandi et al, 2019). The reason that salinity decreased fungal colonization might be attributed to the adverse 5 effects of salinity on photosynthesis, along with a reduction in carbon supply”

The above statements seem contradictory to the claim below on page 5, para 5 – line 1-4: “Our findings showed that P. indica, AMF and their co-inoculation symbiosis mitigated the inhibitory effect of salinity on the photosynthetic capability of peppermint”

Last para – line 1-2: “The endophytic fungus can mitigate the toxic influence of salinity on chloroplast and chlorophyll by releasing cytokinin-like substances which maintain the relative stability of the organelle” – how can the fungi mitigate the toxic influence, when the fungi themselves are negatively affected by salinity?

“Nevertheless, endophytic fungi symbiosis mitigated the inhibitory effect of salt stress on the mineral uptake. It has been well documented that endophytic fungi can prevent the toxicity of Na+ in aerial parts of plants through accumulation of Na+ ions in fungal cytosol, hyphae wall and vesicles” – more explanations are required for this argument, especially when the fungi are harmfully affected by salt stress

• Overall, the researchers have observed evidence for synergistic effect Pirimorphospora and AM in alleviating salt stress in peppermint; however, their arguments need more explanations to avoid self-contradictory appearance as they found salt stress also negatively affecting the plant metabolism as well as fungal colonization. Although, they provide evidences of previous positive findings of synergistic influence of AM and Pirimorphospora in certain plants, they need to report findings showing no such influences as well ( for example: DOI: 10.1080/01904160903435409 )

Reviewer #2: The study applied single P. indica, AMF and co-inoculation of these two endophytic fungi to investigate the potential of enhancing the ability of peppermint to alleviate the negative effect of the salinity stress. Therefore, they are sure interesting and meaningful. However, there are some issues to be addressed before the manuscript is ready for publication. Some language mistakes need to be revised and re-checked. Below are the few comments that can provide the glimpse of flaws in the manuscript.

Q 1 [Introduction] Piriformospora indica, also named as Sebacinales indica, the author needs to add it.

Q 2 [Materials and Methods] Give more specific info on inoculum: e.g. mycelia mass or chlamydospores per ml of P. indica and AMF. And provide the method or reference of fungi inoculation and detection.

Q 3 [Results] This study has many figure, but author doesn’t or good explain the result of the figure.

Q 4 [Discussion] The conclusions drawn by the authors that P. indica, AMF and their co-inoculation symbiosis mitigated the inhibitory effect of salinity on the photosynthetic capability of peppermint is not really supported by those parameters. The authors mention that higher photosynthetic capability attributed to the better physiological status. Why the author not measure the ROS scavengers related enzyme activity to support the point? Kindly explain.

Q 5 The Fig. 2 was not clear, please replace it and provide the scale or microscope magnification. The microscopic observation pictures of the control plant roots are also need provided. “chlamidospore” probably “chlamydospore”? Please check it.

Q 6 Fig. 3, add the abbreviation of Pi. Give the test sample number of root colonization of each group.

Q 7 All the column figures in this study have no error bars, please check/add it.

Q 8 Fig. 5 (a-c) different with other column figure like Fig. 4, please check/replace it.

Q 9 Fig. 6, “AMF* Pi” or “Pi *AMF”? “PI” or “Pi”? Same with Fig. 7, 8, 10, 12 and 13, keep one unified format. please check/replace it.

Q 10 I suggest the authors provide an intuitive picture of each treatment group sample peppermint to show the result of this study.

Here two excellent publications on symbiosis of P. indica and Mentha piperita should be read in this study to give some reference. Dolatabadi, H.K., Goltapeh, E.M., Moieni, A., Varma, A., 2012. Evaluation of different densities of auxin and endophytic fungi (Piriformospora indica and Sebacina vermifera) on Mentha piperita and Thymus vulgaris growth. Afr J Biotechnol, 11: 1644–1650.

Dolatabadi, H.K., Goltapeh, E.M., Safari, M., Golafaie, T.P., 2017. Potential effect of Piriformospora indica on plant growth and essential oil yield in Mentha piperita. Plant Pathol Quar, 7: 96–104.

Overall, the MS needs to be comprehensively revised both in respects of writing and analysis/interpretation of results.

Reviewer #3: Introduction:

1st Para- line 1:- ‘Which its’ can be written as ‘and its’

2nd Para- line 1:- Yield is negatively affected (Kindly add references)

3rd Para- 2nd last line:- ‘Quercus ilex’ should be italics

Material and Methods

Growth Conditions:-

1st para- line 3:- ‘Three replications’; always use minimum five replications

Line 5:- Kindly mention the spore counts of P indica and AMF

Line 6:- Kindly specify the ratio of Caspian sea water and distilled water

Physiological parameters:

Line 3-5:- Kindly elaborate the methods used for analysis

Results:

1st Para- 2nd last line:- Kindly confirm the synergistic association between P indica & AMF through plating techniques.

6. PLOS authors have the option to publish the peer review history of their article (what does this mean?). If published, this will include your full peer review and any attached files.

Reviewer #1: **Yes: **Prof Joseph George Ray

Reviewer #2: No

Reviewer #3: **Yes: **Dr. Md. Nafe Aziz

---

## [Author Response · Author response to Decision Letter 0]

25 Mar 2021

Prof. Mayank Gururani

Editor-in-Chief

PLOS ONE Journal 

Dear Prof., Mayank Gururani

Attached please find our revised manuscript entitled: Does co-inoculation of mycorrhiza and Piriformospora indica fungi enhance the efficiency of Chlorophyll fluorescence and essential oil composition in peppermint under irrigation with saline water from the Caspian Sea? (No.: PONE-D-21-02576) for publication in PLOS ONE Journal.

Thank you for giving us this opportunity to revise and resubmit our manuscript. We appreciate your comments and the suggestions made by referees to improve the manuscript. We responded to all of the reviewers' suggestions. 

We hope that after these enhancements the manuscript can now be accepted for publication; however, we are certainly willing to consider further changes if necessary.

Yours faithfully,

Masoumeh Khalvandi

In behalf authors

Response to Review Comments

(Manuscript Number: PONE-D-21-02576)

Response to Reviewer #1 Comments

Reviewer point #1: Page 1: Introduction: line 1-2: “which its demand has grown dramatically in recent decades” – may be corrected as: ‘the demand of which has grown dramatically in recent decades’

Author response #1: the correction has been made in the revised manuscript.

Reviewer point #2: Line 3: “It is evident that peppermint essential oil” – may be modified as: ‘it is evident that the essential oil’

Author response #2: “It is evident that peppermint essential oil” was modified to ‘it is evident that the essential oil’

Reviewer point #3: Line 5-6: “as well as the reactive oxygen species (ROS) scavenging activity because of its valuable constituents” – ‘as well as scavenging of reactive oxygen species (ROS)’

Author response #3: This has been corrected in the revised manuscript.

Reviewer point #4: 2nd Para – line 1: - avoid ‘Nowadays’; the sentence may begin as ‘The sustainability …’

Author response #4: (Nowadays) was removed in the introduction.

Reviewer point #5: Line 5: no need of repeating – ‘reactive oxygen species (ROS)’, better put the short form only – ROS because the same is already defined above

Author response #5: done

Reviewer point #6: Line 7: “stresses; it can show’ – ‘stresses, because the same can show’

Author response #6: “stresses; it can show’ was modified to ‘stresses, because the same can show’

Reviewer point #7: Page 3: 2nd para – 2nd line: not ‘sever’ – but ‘severe’

Author response #7: ‘sever’ was changed to ‘severe’

Reviewer point #8: Page 4: para 3- line 2: not ‘fungi symbiosis’ – ‘fungal symbiosis’

Author response #8: the correction has been made in the revised manuscript. ‘fungi symbiosis’ was modified to ‘fungal symbiosis’

Reviewer point #9: Discussion: Page 4-5: line 1-4: “In the present study, the root colonization with P. indica and AMF were remarkably reduced by increasing salinity (6 to 9 dSm−1). Similar observations were reported in other plants (Wu et al., 2010; Hadian-Deljou, et al, 2020; khalvandi et al, 2019). The reason that salinity decreased fungal colonization might be attributed to the adverse 5 effects of salinity on photosynthesis, along with a reduction in carbon supply” The above statements seem contradictory to the claim below on page 5, para 5 – line 1-4: “Our findings showed that P. indica, AMF and their co-inoculation symbiosis mitigated the inhibitory effect of salinity on the photosynthetic capability of peppermint”

Reviewer point #10: Last para – line 1-2: “The endophytic fungus can mitigate the toxic influence of salinity on chloroplast and chlorophyll by releasing cytokinin-like substances which maintain the relative stability of the organelle” – how can the fungi mitigate the toxic influence, when the fungi themselves are negatively affected by salinity?

“Nevertheless, endophytic fungi symbiosis mitigated the inhibitory effect of salt stress on the mineral uptake. It has been well documented that endophytic fungi can prevent the toxicity of Na+ in aerial parts of plants through accumulation of Na+ ions in fungal cytosol, hyphae wall and vesicles” – more explanations are required for this argument, especially when the fungi are harmfully affected by salt stress

Author response #9 & 10: 

In this experiment, symbiotic relationship with fungi under no stress and mild stress conditions had a positive significant effect on the examined traits. On the other hand, we observed negative effects of high salinity levels on colonization and physiological parameters of peppermint plants. Although, fungi themselves are negatively affected by salinity, but the results showed that compared to the control plants, symbiotic relationship with these fungi improved the photosynthetic apparatus function. Based on the reports of some researchers, these changes in the presence of fungi and under severe salinity stress can be due to:

The participating AMF induces expression of genes involved in N+ extrusion to the soil solution, K+ acquisition (by phloem loading and unloading), and release into the xylem, therefore maintaining a favorable Na: K ratio. Colonization by AMF differentially affects the expression of the plasma membrane and tonoplast aquaporins (PIPs and TIPs), which consequently improves the water status of the plant. The formation of AM (arbuscular mycorrhiza) surges the capacity of the plant to mend photosystem-II (PSII) and boosts the quantum efficiency of PSII under salt stress conditions by mounting the transcript levels of chloroplast genes encoding antenna proteins involved in the transfer of excitation energy. Furthermore, AM-induced interplay of phytohormones, including strigolactones, abscisic acid, gibberellic acid, salicylic acid, and jasmonic acid have also been associated with the salt tolerance mechanism (Heikham Evelin et al, 2019).

Reviewer point #11: Overall, the researchers have observed evidence for synergistic effect Pirimorphospora and AM in alleviating salt stress in peppermint; however, their arguments need more explanations to avoid self-contradictory appearance as they found salt stress also negatively affecting the plant metabolism as well as fungal colonization. Although, they provide evidences of previous positive findings of synergistic influence of AM and Pirimorphospora in certain plants, they need to report findings showing no such influences as well ( for example: DOI: 10.1080/01904160903435409 )

Author response #11: The correction was made in Conclusion section.

Response to Reviewer #2 Comments

Reviewer point #1: [Introduction] Piriformospora indica, also named as Sebacinales indica, the author needs to add it.

Author response #1: "Sebacinales indica" was added to the text

Reviewer point #2: [Materials and Methods] Give more specific info on inoculum: e.g. mycelia mass or chlamydospores per ml of P. indica and AMF. And provide the method or reference of fungi inoculation and detection.

Author response #2: The methods of fungi inoculation and amount of AMF and P. indica spores were added to the Materials and Methods.

Reviewer point #3: [Results] This study has many figure, but author doesn’t or good explain the result of the figure.

Author response #3: Thank you for pointing this out. The reviewer is correct, we checked all figures and we found in Fv/fm some mistake was made because of careless writing. This has been corrected in the revised manuscript.

Also, Analysis of variance was added to the Results.

Figures 7, 10, and 12 were referred to the text.

Reviewer point #4: [Discussion] The conclusions drawn by the authors that P. indica, AMF and their co-inoculation symbiosis mitigated the inhibitory effect of salinity on the photosynthetic capability of peppermint is not really supported by those parameters. The authors mention that higher photosynthetic capability attributed to the better physiological status. Why the author not measure the ROS scavengers related enzyme activity to support the point? Kindly explain.

Author response #4: Thank you for your suggestion. We studied the antioxidant enzymes, such as catalase and superoxide dismutase, and we have considered it for another article. But if necessary, we can also represent them.

Reviewer point #5: The Fig. 2 was not clear, please replace it and provide the scale or microscope magnification. The microscopic observation pictures of the control plant roots are also need provided. “chlamidospore” probably “chlamydospore”? Please check it.

Author response #5: the scale or microscope magnification was (10- 40X). the scale was added to the pictures. We separated the pictures to make the pictures clearer.

“chlamidospore” was modified to “chlamydospore

Reviewer point #6: Fig. 3, add the abbreviation of Pi. Give the test sample number of root colonization of each group.

Author response #6: The correction was done in text.

Reviewer point #7: All the column figures in this study have no error bars, please check/add it.

Author response #7: Error bars was added to the all figures.

Reviewer point #8: Fig. 5 (a-c) different with other column figure like Fig. 4, please check/replace it.

Author response #8: Thank you for pointing this out. The reviewer is correct, this mistake was made because of using the wrong way of lettering Means. We learned from one of our colleagues that there is a more accurate way to Lettering means. then, we did the Lettering method through the SAS program itself, and all figures have been corrected.

Reviewer point #9: Fig. 6, “AMF* Pi” or “Pi *AMF”? “PI” or “Pi”? Same with Fig. 7, 8, 10, 12 and 13, keep one unified format. please check/replace it.

Author response #9: The reviewer is correct; the correction has been made in the revised manuscript.

Reviewer point #10: I suggest the authors provide an intuitive picture of each treatment group sample peppermint to show the result of this study.

Author response #10: Thank you for your valuable suggestion. This research has already been done and we only have some pictures from the time of planting that we present.

Reviewer point #11: Here two excellent publications on symbiosis of P. indica and Mentha piperita should be read in this study to give some reference. Dolatabadi, H.K., Goltapeh, E.M., Moieni, A., Varma, A., 2012. Evaluation of different densities of auxin and endophytic fungi (Piriformospora indica and Sebacina vermifera) on Mentha piperita and Thymus vulgaris growth. Afr J Biotechnol, 11: 1644–1650.

Dolatabadi, H.K., Goltapeh, E.M., Safari, M., Golafaie, T.P., 2017. Potential effect of Piriformospora indica on plant growth and essential oil yield in Mentha piperita. Plant Pathol Quar, 7: 96–104.

Author response #11: These references were added to the revised manuscript.

Response to Reviewer #3 Comments

Reviewer point #1: 1st Para- line 1:- ‘Which its’ can be written as ‘and its’

Author response #1: ‘Which its’ was modified to ‘and its’

Reviewer point #2: 2nd Para- line 1:- Yield is negatively affected (Kindly add references)

Author response #2: The appropriate reference was added to "Yield is negatively affected"

Reviewer point #3: 3rd Para- 2nd last line:- ‘Quercus ilex’ should be italics

Author response #3: The correction was done in text.

Reviewer point #4: 1st para- line 3:- ‘Three replications’; always use minimum five replications

Author response #4: Thanks to the referee’s suggestion, definitely conducting the experiment with five replications would be more accurate. However, we found authoritative articles that tested their experiment with three repetitions, such as:

1. Bharti, N., Pandey, S. S., Barnawal, D., Patel, V. K., & Kalra, A. (2016). Plant growth promoting rhizobacteria Dietzia natronolimnaea modulates the expression of stress responsive genes providing protection of wheat from salinity stress. Scientific reports, 6(1), 1-16.

2. Gururani, M. A., Venkatesh, J., Ghosh, R., Strasser, R. J., Ponpandian, L. N., & Bae, H. (2018). Chlorophyll-a fluorescence evaluation of PEG-induced osmotic stress on PSII activity in Arabidopsis plants expressing SIP1. Plant Biosystems-An International Journal Dealing with all Aspects of Plant Biology, 152(5), 945-952.

3. Li, X., Zhao, C., Zhang, T., Wang, G., Amombo, E., Xie, Y., & Fu, J. (2021). Exogenous Aspergillus aculeatus Enhances Drought and Heat Tolerance of Perennial Ryegrass. Frontiers in Microbiology, 12, 307.

4. Varghese, N., Alyammahi, O., Nasreddine, S., Alhassani, A., & Gururani, M. A. (2019). Melatonin positively influences the photosynthetic machinery and antioxidant system of Avena sativa during salinity stress. Plants, 8(12), 610.

5. Yamane, K., Oi, T., Enomoto, S., Nakao, T., Arai, S., Miyake, H., & Taniguchi, M. (2018). Three‐dimensional ultrastructure of chloroplast pockets formed under salinity stress. Plant, cell & environment, 41(3), 563-575.

6. Ji, J., Yue, J., Xie, T., Chen, W., Du, C., Chang, E., ... & Shi, S. (2018). Roles of γ-aminobutyric acid on salinity-responsive genes at transcriptomic level in poplar: involving in abscisic acid and ethylene-signalling pathways. Planta, 248(3), 675-690.

Reviewer point #5: Line 5:- Kindly mention the spore counts of P indica and AMF

Author response #5: 10 g of Arbuscular mycorrhizal inoculum (with a density of 120 active spores per gram) and 10 ml of Piriformospora indica suspension (1× 109) were added to the pots.

Reviewer point #6: Line 6:- Kindly specify the ratio of Caspian sea water and distilled water

Author response #6: In order to reach each level of salinity stress: determined amount of the Caspian Sea water and distilled water were mixed and then the desired salinity was determined by EC meter. 

Reviewer point #7: Physiological parameters: Line 3-5:- Kindly elaborate the methods used for analysis

Author response #7: The methods used for analysis were added to the Materials and Methods section.

Chlorophyll fluorescence parameters were measured in the last fully developed leaf by using pulse amplitude modulated fluorometer (PAM-2500, Walz, Germany). First, leaves were placed in darkness for 30 minutes using specific leaf clamps. The samples were exposed to low-intensity light [< 0.1 μmol (photon) m−2 s−1, red light]. Then, a saturating light pulse [> 8,000 μmol (photon) m−2 s−1, white light) was turned on for 1 s (one pulse). The minimum fluorescence (Fo) and maximum fluorescence (Fm) were determined in dark-adapted leaves. The variable fluorescence (Fv) and maximum quantum photosystem II efficiency (Fv/Fm) were evaluated based on Equations 1 and 2. and the effective photochemical quantum efficiency II [ФPSII], electron transfer rate [ETR, μmol(electron) m–2 s–1), non-photochemical quenching (NPQ), the quantum yield of regulated energy dissipation (YNPQ), and Quantum yield of non-regulated energy dissipation in PSII (Y (NO)) were calculated by using Equations 3 and 6 (Khanghahi et al, 2012).

1) Fv= Fm- Fo

2) Fv/Fm= (Fm- Fo)/ Fm

3) ФPSII= (Fm´ − F)/Fm´

4) NPQ = (Fm/Fm') – 1

5) Y(NPQ) = (F/Fm') – (F/Fm)

6) Y(NO) = F/Fm

F= steady-state fluorescenc

Fm'= maximum fluorescence measured in light-exposed leaf samples.

Reviewer point #8: Results: 1st Para- 2nd last line:- Kindly confirm the synergistic association between P indica & AMF through plating techniques.

Author response #8: Thank you for your valuable suggestion, we definitely investigate the synergistic association between P indica & AMF through plating techniques in our future research; however, this research has already been conducted and done; thus, we can't add a new experiment.

---

## [Decision Letter · Decision Letter 1]

20 Apr 2021

PONE-D-21-02576R1

Does co-inoculation of mycorrhiza and Piriformospora indica fungi enhance the efficiency of Chlorophyll fluorescence and essential oil composition in peppermint under irrigation with saline water from the Caspian Sea?

PLOS ONE

Dear Dr. Khalvandi,

Thank you for submitting your manuscript to PLOS ONE. After careful consideration, we feel that it has merit but does not fully meet PLOS ONE’s publication criteria as it currently stands. Therefore, we invite you to submit a revised version of the manuscript that addresses the points raised during the review process.

We look forward to receiving your revised manuscript.

Kind regards,

Mayank Gururani

Academic Editor

PLOS ONE

Journal Requirements:

Reviewers' comments:

Reviewer's Responses to Questions

**Comments to the Author**

1. If the authors have adequately addressed your comments raised in a previous round of review and you feel that this manuscript is now acceptable for publication, you may indicate that here to bypass the “Comments to the Author” section, enter your conflict of interest statement in the “Confidential to Editor” section, and submit your "Accept" recommendation.

Reviewer #1: (No Response)

Reviewer #3: All comments have been addressed

2. Is the manuscript technically sound, and do the data support the conclusions?

Reviewer #1: Yes

Reviewer #3: Yes

3. Has the statistical analysis been performed appropriately and rigorously? 

Reviewer #1: (No Response)

Reviewer #3: Yes

4. Have the authors made all data underlying the findings in their manuscript fully available?

Reviewer #1: Yes

Reviewer #3: Yes

5. Is the manuscript presented in an intelligible fashion and written in standard English?

Reviewer #1: No

Reviewer #3: Yes

6. Review Comments to the Author

Reviewer #1: General comments: The article is improved a lot after the revision, but the authors were not thorough in a critical review of the entire text. Some minor corrections in the abstract, introduction and materials & methods part of the article are pointed out below.

But the discussion part needs thorough revision. The authors discuss many physiological effects in plants concerning salt stress based on previous literature but not specific to their findings. Discussion in a research paper is meant to discuss the results and not the previous literature. Some contradictions are also in their argument. They must thoroughly revise the discussion. Let them go through the specific comments pointed out below while reviewing the discussion. If they limit the discussion to their actual findings, the entire discussion can be summarized to 1/3rd of what is presented.

Specific comments

1. Abstract: First sentence of the abstracts seems complex and needs to be simplified

2. Introduction – 2nd paragraph – 3rd line: ‘Photosynthesis process’ – please delete ‘process’

3. Introduction – 4th paragraph – Line No.12: ‘which are all pharmaceutical useful’ – please modify as ‘pharmaceutically useful’

4. Material and Methods: 1st paragraph – 2nd line: ‘This research was carried out in a factorial experiment in a….’ – may be modified as: ‘A factorial experiment was carried out in a....’

5. Discussion – 9th paragraph: ‘Several reports have reported a direct correlation between nutrient imbalance and the simultaneous decline in PSII function in salt-stressed plants’ - the sentence may be modified as ‘Several reports suggest a direct correlation between nutrient imbalance and the simultaneous decline in PSII function in salt-stressed plants’ – which are these reports? Please refer to the reports; the reference of Yang et al. (2021) is not sufficient to substantiate the argument of ‘several reports’; otherwise, modify the sentence.

9th paragraph – line 7: ‘fungi symbiosis’ – may be modified as fungal symbiosis

9th paragraph: lines 6-7: ‘Nevertheless, endophytic fungi symbiosis mitigated the inhibitory effect of salt stress on the mineral uptake’ – please mention the evidence which the authors have presented as results in their study to substantiate this argument; otherwise delete this speculative argument. (the first sentence of this paragraph is ‘The results of this study showed that severe salinity stress remarkably reduced P and K+ uptake because of the high rate of Na+ absorption’, is contradictory to this argument.

Line 11: ‘osmotic stress)’ – delete the bracket

10th paragraph: lines 5-7: ‘In agreement with the present results, Xu et al. (2016) reported that the positive effect of AMF on Fv/Fm increasing in maize seedlings can be attributed to the absorption of mineral nutrients, activating mediated genes and sink stimulation’ – how can it be in agreement with, especially when the authors argue in the 9th paragraph that ‘The results of this study showed that severe salinity stress remarkably reduced P and K+ uptake because of the high rate of Na+ absorption’ ? - please note the contradiction and revise

I think the 11th and 12th paragraph are unnecessary as these do not discuss any specific findings of the researchers.

Overall, the discussion need to be thoroughly revised

Reviewer #3: The manuscript describe a technically sound piece of scientific research. Authors have adequately addressed the comments raised in a previous round of review and this manuscript is now acceptable for publication

7. PLOS authors have the option to publish the peer review history of their article (what does this mean?). If published, this will include your full peer review and any attached files.

Reviewer #1: **Yes: **Joseph George Ray

Reviewer #3: **Yes: **Md Nafe Aziz

---

## [Author Response · Author response to Decision Letter 1]

5 May 2021

Prof. Mayank Gururani

Editor-in-Chief

PLOS ONE Journal 

Dear Prof., Mayank Gururani

Attached please find our revised manuscript entitled: Does co-inoculation of mycorrhiza and Piriformospora indica fungi enhance the efficiency of Chlorophyll fluorescence and essential oil composition in peppermint under irrigation with saline water from the Caspian Sea? (No.: PONE-D-21-02576) for publication in PLOS ONE Journal.

Thank you for giving us this opportunity to revise and resubmit our manuscript. We appreciate your comments and the suggestions made by referees to improve the manuscript. We responded to all of the reviewers' suggestions. 

We hope that after these enhancements the manuscript can now be accepted for publication; however, we are certainly willing to consider further changes if necessary.

Yours faithfully,

Masoumeh Khalvandi

In behalf authors

Response to Review Comments

(Manuscript Number: PONE-D-21-02576)

Response to Reviewer #1 Comments

Reviewer point #1: If they limit the discussion to their actual findings, the entire discussion can be summarized to 1/3rd of what is presented. 

Author response #1: The length of the discussion part has been reduced. And these sentences were removed in the discussion part.

• The reason that salinity decreased fungal colonization might be attributed to the adverse effects of salinity on photosynthesis, along with a reduction in carbon supply from plant to endophytic fungus. This can also be supported by the fact that salinity induced changes in morphological features of fungal hyphae. Besides,

• These results are similar to what has been recorded in wheat (lyas et al, 2020), and maize (Wang et al, 2020).

• inhibition of QA re-oxidation, and degradation of D1 and D2 proteins in the PSII which

• Salinity stress probably saturates the electron transport chain that leads to accumulation of protons, followed by an increase in the amount of NPQ. The increase in NPQ, which is actually an indicator of heat loss in the processes of the excessive energy removal, shows the high capacity of the xanthophyll cycle (Nilkens et al., 2010; Murchie and Lawson, 2013).

• Higher values of photochemical activity of PSII in plants which have a symbiotic relationship with fungi are mainly associated with: the increased density of photosynthetic units, the improved reaction center activity, enhancing the plants’ ability to use light energy, and facilitating the electron transport to NADP (Xu et al, 2016; Moreira et al. 2015; Zhang et al, 2018; Shahabivand et al, 2017; Ghorbani et al, 2018; Shahabivand et al, 2012). Furthermore, microbial inoculation can alleviate the adverse impacts of salinity on the photochemical activity of PSII through expression of the genes encoding D1 and D2 proteins, improving water absorption, modulating membrane damage, and improving chlorophyll content (Chen et al, 2017).

• AMF has been reported to regulate root permeability by doubling the expression of the LSPLP1 gene, the strategy which leads to a higher tolerance to osmotic stress) (Heikham Evelin et al, 2019).

• In addition to what is mentioned above, salinity suppresses the biosynthesis of essential oil. The reason for the reduction in the synthesis of essential oils can be related to the low availability of assimilates as a result of reduced photosynthetic activity and inactivation of the PSII reaction center (Zuo et al, 2017). However, even when photosynthesis assimilation approaches zero (due to stomatal restriction, damage to chlorophyll, and chloroplast membrane system under severe stress conditions) monoterpenes biosynthesis still occurs due to alternative carbon sources allocation (Lavoir et al, 2009). Naturally, monoterpenes are biosynthesized through MEP/ DOXP pathway and are fueled by a constant supply of substrates, high amounts of acetyl-CoA, and ATP and NADPH+H cofactors (Niinemets et al., 2002, Behn et al, 2010). This can be the reason for the tight link between methyl erythritol-4-osphate (MEP) and photosynthesis, as the Calvin cycle in chloroplasts is the main carbon sources for terpenoid biosynthesis and the MEP pathway is a place for reducing excess electron flow in the photosynthetic electron transfer chain (Valifard et al, 2018). 

• It is believed that the monoterpene pattern is mostly determined by NADPH+H+ -dependent reductive interconversions, like a reduction of menthone to menthol (Behn et al, 2010). These modifications in the biosynthesis of monoterpenes can be linked to alterations in the photosynthetic electron transfer chain, the production of photosynthetic NDPH2 (Maffei and Codignola, 1990), and oxidation of monoterpenes by ROS to generate oxygenated forms under environmental stress (Zuo et al, 2017).

These references were also removed in the revised manuscript.

1. Agron. 10, 160–162.Wu, M., Wei, Q., Xu, L., Li, H., Oelmüller, R., & Zhang, W. (2018). Piriformospora indica enhances phosphorus absorption by stimulating acid phosphatase activities and organic acid accumulation in Brassica napus. Plant and Soil, 432(1-2), 333-344.

2. Behn, H., Albert, A., Marx, F., Noga, G., & Ulbrich, A. (2010). Ultraviolet-B and photosynthetically active radiation interactively affect yield and pattern of monoterpenes in leaves of peppermint (Mentha× piperita L.). Journal of agricultural and food chemistry, 58(12), 7361-7367.

3. Flexas, J., Escalona, J. M., & Medrano, H. (1999). Water stress induces different levels of photosynthesis and electron transport rate regulation in grapevines. Plant, Cell & Environment, 22(1), 39-48.

4. lyas, N., Mazhar, R., Yasmin, H., Khan, W., Iqbal, S., Enshasy, H. E., & Dailin, D. J. (2020). Rhizobacteria isolated from saline soil induce systemic tolerance in wheat (Triticum aestivum L.) against salinity stress. Agronomy, 10(7), 989.

5. Lavoir, A. V., Staudt, M., Schnitzler, J. P., Landais, D., Massol, F., Rocheteau, A., ... & Rambal, S. (2009). Drought reduced monoterpene emissions from the evergreen Mediterranean oak Quercus ilex: results from a throughfall displacement experiment. Biogeosciences, 6(7), 1167-1180.

6. Maffei, M., & Codignola, A. (1990). Photosynthesis, photorespiration and herbicide effect on terpene production in peppermint (Mentha piperita L.). Journal of Essential Oil Research, 2(6), 275-286.

7. Sebastian, A., & Prasad, M. N. V. (2019). Photosynthetic light reactions in Oryza sativa L. under Cd stress: Influence of iron, calcium, and zinc supplements. The EuroBiotech Journal. 3(4): 175-181.

8. Zuo, Z., Wang, B., Ying, B., Zhou, L., & Zhang, R. (2017). Monoterpene emissions contribute to thermotolerance in Cinnamomum camphora. Trees, 31(6), 1759-1771.

Reviewer point #2: Abstract: First sentence of the abstracts seems complex and needs to be simplified 

Author response #2: The correction was done in Abstract.

Reviewer point #3: Introduction – 2nd paragraph – 3rd line: ‘Photosynthesis process’ – please delete ‘process’ 

Author response #3: process was deleted in ‘Photosynthesis process’

Reviewer point #4: Introduction – 4th paragraph – Line No.12: ‘which are all pharmaceutical useful’ – please modify as ‘pharmaceutically useful’ 

Author response #4: ‘which are all pharmaceutical useful’ was modified to ‘pharmaceutically useful’

Reviewer point #5: Material and Methods: 1st paragraph – 2nd line: ‘This research was carried out in a factorial experiment in a….’ – may be modified as: ‘A factorial experiment was carried out in a....’ 

Author response #5: ‘This research was carried out in a factorial experiment in a….’ was modified to ‘A factorial experiment was carried out in a....’ 

Reviewer point #6: Discussion – 9th paragraph: ‘Several reports have reported a direct correlation between nutrient imbalance and the simultaneous decline in PSII function in salt-stressed plants’ - the sentence may be modified as ‘Several reports suggest a direct correlation between nutrient imbalance and the simultaneous decline in PSII function in salt-stressed plants’

Author response #6: The correction has been made in the revised manuscript.

Reviewer point #7: which are these reports? Please refer to the reports; the reference of Yang et al. (2021) is not sufficient to substantiate the argument of ‘several reports’; otherwise, modify the sentence.

Author response #7: Thank you for pointing this out. The following articles were added to the references.

• Loudari, A., Benadis, C., Naciri, R., Soulaimani, A., Zeroual, Y., Gharous, M. E., Kalaji, H M., & Oukarroum, A. (2020). Salt stress affects mineral nutrition in shoots and roots and chlorophyll a fluorescence of tomato plants grown in hydroponic culture. Journal of Plant Interactions, 15(1), 398-405.

• Qu, C., Liu, C., Gong, X., Li, C., Hong, M., Wang, L., & Hong, F. (2012). Impairment of maize seedling photosynthesis caused by a combination of potassium deficiency and salt stress. Environmental and Experimental Botany, 75, 134-141

Reviewer point #8: 9th paragraph – line 7: ‘fungi symbiosis’ – may be modified as fungal symbiosis 

Author response #8: ‘fungi symbiosis’ was modified to ‘fungal symbiosis’ 

Reviewer point #9: 9th paragraph: lines 6-7: ‘Nevertheless, endophytic fungi symbiosis mitigated the inhibitory effect of salt stress on the mineral uptake’ – please mention the evidence which the authors have presented as results in their study to substantiate this argument; otherwise delete this speculative argument. (the first sentence of this paragraph is ‘The results of this study showed that severe salinity stress remarkably reduced P and K+ uptake because of the high rate of Na+ absorption’, is contradictory to this argument.

Author response #9: ‘Nevertheless, endophytic fungi symbiosis mitigated the inhibitory effect of salt stress on the mineral uptake’ was modified to ‘Nevertheless, in inoculated plant endophytic fungal symbiosis increased P and K+ content (Figure. 6a) and decreased Na+ uptake in peppermint leaves (Figure. 4c).’

In order to avoid ambiguity and contradiction, ‘in non-inoculated plant’ and ‘in inoculated plant’ were added to the text of this paragraph.

Reviewer point #10: Line 11: ‘osmotic stress)’ – delete the bracket

Author response #10: done

Reviewer point #11: 10th paragraph: lines 5-7: ‘In agreement with the present results, Xu et al. (2016) reported that the positive effect of AMF on Fv/Fm increasing in maize seedlings can be attributed to the absorption of mineral nutrients, activating mediated genes and sink stimulation’ – how can it be in agreement with, especially when the authors argue in the 9th paragraph that ‘The results of this study showed that severe salinity stress remarkably reduced P and K+ uptake because of the high rate of Na+ absorption’ ? - please note the contradiction and revise

Author response #11: 

In order to avoid ambiguity and contradiction, ‘The results of this study showed that severe salinity stress remarkably reduced P and K+ uptake because of the high rate of Na+ absorption’ was modified to ‘The results of this study showed that severe salinity stress remarkably reduced P and K+ uptake in non-inoculated plant because of the high rate of Na+ absorption.’

Thank you for your valuable suggestion. After reviewing articles in various journals, we found that they also reported such contradictions, and in all of them, although the rate of colonization and formation of fungal hyphae and spores decreased under salinity stress, its presence in the plant roots was able to reduce the negative effects of salinity stress on the plant and improves the amount of some plant parameters.

• Klinsukon, C., Lumyong, S., Kuyper, T. W., & Boonlue, S. (2021). Colonization by arbuscular mycorrhizal fungi improves salinity tolerance of eucalyptus (Eucalyptus camaldulensis) seedlings. Scientific Reports, 11(1), 1-10.

• Zai, X. M., Fan, J. J., Hao, Z. P., Liu, X. M., & Zhang, W. X. (2021). Effect of co-inoculation with arbuscular mycorrhizal fungi and phosphate solubilizing fungi on nutrient uptake and photosynthesis of beach palm under salt stress environment. Scientific Reports, 11(1), 1-11.

• Parvin, S., Van Geel, M., Yeasmin, T., Verbruggen, E., & Honnay, O. (2020). Effects of single and multiple species inocula of arbuscular mycorrhizal fungi on the salinity tolerance of a Bangladeshi rice (Oryza sativa L.) cultivar. Mycorrhiza, 30(4), 431-444.

• Kaya, C., Ashraf, M., Sonmez, O., Aydemir, S., Tuna, A. L., & Cullu, M. A. (2009). The influence of arbuscular mycorrhizal colonisation on key growth parameters and fruit yield of pepper plants grown at high salinity. Scientia horticulturae, 121(1), 1-6.

• Jahromi, F., Aroca, R., Porcel, R., & Ruiz-Lozano, J. M. (2008). Influence of salinity on the in vitro development of Glomus intraradices and on the in vivo physiological and molecular responses of mycorrhizal lettuce plants. Microbial Ecology, 55(1), 45.

Reviewer point #12: I think the 11th and 12th paragraph are unnecessary as these do not discuss any specific findings of the researchers.

Author response #12: These 11th and 12th paragraphs was delated in the revised manuscript.

---

## [Decision Letter · Decision Letter 2]

21 May 2021

PONE-D-21-02576R2

Does co-inoculation of mycorrhiza and Piriformospora indica fungi enhance the efficiency of Chlorophyll fluorescence and essential oil composition in peppermint under irrigation with saline water from the Caspian Sea?

PLOS ONE

Dear Dr. Khalvandi,

Thank you for submitting your manuscript to PLOS ONE. After careful consideration, we feel that it has merit but does not fully meet PLOS ONE’s publication criteria as it currently stands. Therefore, we invite you to submit a revised version of the manuscript that addresses the points raised during the review process.

We look forward to receiving your revised manuscript.

Kind regards,

Mayank Gururani

Academic Editor

PLOS ONE

Journal Requirements:

Reviewers' comments:

Reviewer's Responses to Questions

**Comments to the Author**

1. If the authors have adequately addressed your comments raised in a previous round of review and you feel that this manuscript is now acceptable for publication, you may indicate that here to bypass the “Comments to the Author” section, enter your conflict of interest statement in the “Confidential to Editor” section, and submit your "Accept" recommendation.

Reviewer #1: (No Response)

Reviewer #4: All comments have been addressed

2. Is the manuscript technically sound, and do the data support the conclusions?

Reviewer #1: Yes

Reviewer #4: Yes

3. Has the statistical analysis been performed appropriately and rigorously? 

Reviewer #1: N/A

Reviewer #4: Yes

4. Have the authors made all data underlying the findings in their manuscript fully available?

Reviewer #1: Yes

Reviewer #4: Yes

5. Is the manuscript presented in an intelligible fashion and written in standard English?

Reviewer #1: No

Reviewer #4: Yes

6. Review Comments to the Author

Reviewer #1: I appreciate the improvements done in the submitted revision text, but feel that the authors need to summarize the discussion further. The lengthy and imprecise debate can only tarnish the brilliance of findings in any good research. Therefore, I request the authors to revise the ‘discussion’ to clarify the exact relevance of their findings. Please use the comments given in the ‘discussion’ part of the revised 'manuscript with track changes' as a model for the revision. Please avoid excessive references to previous findings; limit the citations to a maximum of 1-2 most relevant authorities in each case. Please also thoroughly check the entire text before submission of the revision for spelling and grammar; summarize or rephrase sentences to improve clarity wherever necessary. Finally, check the references thoroughly after deletion of unnecessary citations in the text during revision of the text.

Reviewer #4: The authors have adequately addressed the points raised by the previous reviewers and the manuscript can be accepted for publication. However, the article will look more authentic if you could mention the source of both the Piriformospora indica and Arbuscular mycorrhizal fungi used in this work. If available, mention the culture depository details as well.

7. PLOS authors have the option to publish the peer review history of their article (what does this mean?). If published, this will include your full peer review and any attached files.

Reviewer #1: **Yes: **Joseph George Ray

Reviewer #4: No

---

## [Author Response · Author response to Decision Letter 2]

10 Jun 2021

Prof. Mayank Gururani

Editor-in-Chief

PLOS ONE Journal 

Dear Prof., Mayank Gururani

Attached please find our revised manuscript entitled: Does co-inoculation of mycorrhiza and Piriformospora indica fungi enhance the efficiency of Chlorophyll fluorescence and essential oil composition in peppermint under irrigation with saline water from the Caspian Sea? (No.: PONE-D-21-02576) for publication in PLOS ONE Journal.

Thank you for giving us this opportunity to revise and resubmit our manuscript. We appreciate your comments and the suggestions made by referees to improve the manuscript. We responded to all of the reviewers' suggestions. 

We hope that after these enhancements the manuscript can now be accepted for publication; however, we are certainly willing to consider further changes if necessary.

Yours faithfully,

Masoumeh Khalvandi

In behalf authors

Response to Review Comments

(Manuscript Number: PONE-D-21-02576)

Response to Reviewer #1 Comments

Reviewer point #1: The lengthy and imprecise debate can only tarnish the brilliance of findings in any good research. Therefore, I request the authors to revise the ‘discussion’ to clarify the exact relevance of their findings. Please use the comments given in the ‘discussion’ part of the revised 'manuscript with track changes' as a model for the revision. Please avoid excessive references to previous findings; limit the citations to a maximum of 1-2 most relevant authorities in each case. Please also thoroughly check the entire text before submission of the revision for spelling and grammar; summarize or rephrase sentences to improve clarity wherever necessary. Finally, check the references thoroughly after deletion of unnecessary citations in the text during revision of the text.

Author response #1: All requested corrections were made in the revised manuscript. The comments made in the ‘discussion’ part of the 'revised manuscript with track changes' were used as a model for revision. citations were reduced to a maximum of 1-2 most relevant authorities in each case. unnecessary references were deleted. The text was grammatically reviewed and corrected.

The following references have been deleted:

1. Calatayud, A., Roca, D., Martínez, P.F., (2006). Spatial-temporal variations in rose leaves under water stress conditions studied by chlorophyll ﬂuorescence imaging. Plant Physiol. Biochem. 44, 564e573

2. Coban, O., Göktürk Baydar, N., (2016). Brassinosteroid effects on some physical and biochemical properties and secondary metabolite accumulation in peppermint (Mentha piperita L.) under salt stress. Ind Crops Prod. 86, 251-258.

3. Gupta, R., Singh, A., Srivastava, M., Singh, V., Gupta, M. M., & Pandey, R. (2017). Microbial modulation of bacoside A biosynthetic pathway and systemic defense mechanism in Bacopa monnieri under Meloidogyne incognita stress. Scientific reports, 7(1), 1-11.

4. Hasan, M., Ma, F., Prodhan, Z., Li, F., Shen, H., Chen, Y., & Wang, X. (2018). Molecular and physio-biochemical characterization of cotton species for assessing drought stress tolerance. International journal of molecular sciences, 19(9), 2636.

5. Khatri, K., & Rathore, M. S. (2019). Photosystem photochemistry, prompt and delayed fluorescence, photosynthetic responses and electron flow in tobacco under drought and salt stress. Photosynthetica, 57(1), 61-74.

6. Kromdijk, J., Głowacka, K., Leonelli, L., Gabilly, S.T., Iwai, M., Niyogi, K.K., Long, S.P. (2016). Improving photosynthesis and crop productivity by accelerating recovery from photoprotection. Science 354, 857–861.

7. Kumar, G. (2020). Unit 7 Topic: Arbuscular Mycorrhizal Colonization In Plant Roots.

8. Li, D., Liu, H., Qiao, Y., Wang, Y., Cai, Z., Dong, B., Shi, C., Liu, Y., Li, X., and Liu, M. (2013). Effects of elevated CO2 on the growth, seed yield, and water use efficiency of soybean (Glycine max (L.) Merr.) under drought stress. Agricultural Water Management, 129, 105-112.

9. Li, Y., Song, H., Zhou, L., Xu, Z., & Zhou, G. (2019). Tracking chlorophyll fluorescence as an indicator of drought and rewatering across the entire leaf lifespan in a maize field. Agricultural Water Management, 211, 190-201.

10. Loomis, W.D., Corteau, R., 1972. Essential oil biosynthesis. Recent Adv Phytochem. 6, 147-185.

11. McMillen, B.G., Juniper, S., Abbott, L.K., (1998). Inhibition of hyphal growth of a Vesicular arbuscular mycorrhizal fungus in soil containing sodium chloride limits the spread of infection from spores. Soil Biol. Biochem. 30, 1639–1646.

12. Moreira, B. C., Junior, P. P., Jordao, T. C., da Silva, M. D. C. S., Stürmer, S. L., Salomão, L. C. C., ... & Kasuya, M. C. M. (2016). Effect of inoculation of symbiotic fungi on the growth and antioxidant enzymes’ activities in the presence of Fusarium subglutinans f. sp. ananas in pineapple plantlets. Acta Physiologiae Plantarum, 38(10), 1-14.

13. Niinemets, Ü., Hauff, K., Bertin, N., Tenhunen, J. D., Steinbrecher, R., & Seufert, G. (2002). Monoterpene emissions in relation to foliar photosynthetic and structural variables in Mediterranean evergreen Quercus species. New Phytologist, 153(2), 243-256.

14. Nilkens, M., Kress, E., Lambrev, P., Miloslavina, Y., Müller, M., Holzwarth, A.R., Jahns, P. (2010). Identification of a slowly inducible zeaxanthin-dependent component of nonphotochemical quenching of chlorophyll fluorescence generated under steady-state conditions in Arabidopsis.

15. Ormeño, E., Olivier, R., Mévy, J. P., Baldy, V., & Fernandez, C. (2009). Compost may affect volatile and semi-volatile plant emissions through nitrogen supply and chlorophyll fluorescence. Chemosphere, 77(1), 94-104.

16. Rai, M., Acharya, D., Singh, A., Varma, A., (2001). Positive growth responses of the medicinal plants Spilanthes calva and Withania somnifera to inoculation by Piriformospora indica in a field trial. Mycorrhiza. 11, 123-128.

17. Rapparini, F., Llusià, J., & Peñuelas, J. (2008). Effect of arbuscular mycorrhizal (AM) colonization on terpene emission and content of Artemisia annua L. Plant Biology, 10(1), 108-122.

18. Shahabivand, S., Parvaneh, A., & Aliloo, A. A. (2017). Root endophytic fungus Piriformospora indica affected growth, cadmium partitioning and chlorophyll fluorescence of sunflower under cadmium toxicity. Ecotoxicology and environmental safety, 145, 496-502.

19. Sun, C., Johnson, J. M., Cai, D., Sherameti, I., Oelmüller, R., & Lou, B. (2010). Piriformospora indica confers drought tolerance in Chinese cabbage leaves by stimulating antioxidant enzymes, the expression of drought-related genes and the plastid-localized CAS protein. Journal of plant physiology, 167(12), 1009-1017.

20. Tarraf, W., Ruta, C., Cillis, F.D., Tagarelli, A., Tedone, L., Mastro, G.D., (2015). Effects of mycorrhiza on growth and essential oil production in selected aromatic plants. J.

21. Vickers, C. E., Gershenzon, J., Lerdau, M. T., & Loreto, F. (2009). A unified mechanism of action for volatile isoprenoids in plant abiotic stress. Nature chemical biology, 5(5), 283.

22. Tsai, M.L., Wu, C.T., Lin, T.F., Lin, W.C., Huang, Y.C., Yang, C.H., (2013). Chemical composition and biological properties of essential oils of two mint species. Trop. J. Pharm. Res. 12, 577–582.

Reviewer #4: The authors have adequately addressed the points raised by the previous reviewers and the manuscript can be accepted for publication. However, the article will look more authentic if you could mention the source of both the Piriformospora indica and Arbuscular mycorrhizal fungi used in this work. If available, mention the culture depository details as well.

Author response #1: The source of both the Piriformospora indica and Arbuscular mycorrhizal fungi used in this work was added in text.

We obtained both he Piriformospora indica and Arbuscular mycorrhizal fungi from Sari Agricultural Sciences and Natural Resources University. But the main source of these two fungi was as follows: Mycorrhiza fungi inoculum (consisted of spores in a sand and mycorrhizal roots mixture) was prepared from Turan Biotechnology Company, Shahrood, Iran. The Piriformospora indica culture was kindly gifted by Prof. Karl-Heinz Kogel, Institute of Phytopathology and Applied Zoology, University of Giessen, Germany. P. indica was cultured in liquid Kafer’s medium at 24◦C for 10 days (Yaghoubian et al., 2019).

Journal Requirements:

Author response #1: reference list has been corrected in the revised manuscript.

"Arnon, D. I. (1949). Copper enzymes in isolated chloroplasts. Polyphenol oxidase in Beta vulgaris. Plant physiology, 24(1), 1" was replaced with "Hameed, A., Akram, N. A., Saleem, M. H., Ashraf, M., Ahmed, S., Ali, S., ... & Alyemeni, M. N. (2021). Seed treatment with α-tocopherol regulates growth and key physio-biochemical attributes in carrot (Daucus carota l.) plants under water limited regimes. Agronomy, 11(3), 469".

"Hemming, D., (2013). Plant sciences reviews 2012. CABI. 280 pp. (online ISSN 1749-8848)" was replaced with " Sun, Z., Wang, H., Wang, J., Zhou, L., Yang, P., (2014). Chemical composition and anti-inflammatory, cytotoxic and antioxidant activities of essential oil from Leaves of Mentha piperita grown in china. PLoS One. 12, e114767."

"Bertamini, M., Faralli, M., Varotto, C., Grando, M. S., & Cappellin, L. (2021). Leaf Monoterpene Emission Limits Photosynthetic Downregulation under Heat Stress in Field-Grown Grapevine. Plants, 10(1), 181" was corrected.

---

## [Decision Letter · Decision Letter 3]

21 Jun 2021

Does co-inoculation of mycorrhiza and Piriformospora indica fungi enhance the efficiency of Chlorophyll fluorescence and essential oil composition in peppermint under irrigation with saline water from the Caspian Sea?

PONE-D-21-02576R3

Dear Dr. Khalvandi,

We’re pleased to inform you that your manuscript has been judged scientifically suitable for publication and will be formally accepted for publication once it meets all outstanding technical requirements.

Kind regards,

Mayank Gururani

Academic Editor

PLOS ONE

Additional Editor Comments (optional):

Reviewers' comments:

Reviewer's Responses to Questions

**Comments to the Author**

1. If the authors have adequately addressed your comments raised in a previous round of review and you feel that this manuscript is now acceptable for publication, you may indicate that here to bypass the “Comments to the Author” section, enter your conflict of interest statement in the “Confidential to Editor” section, and submit your "Accept" recommendation.

Reviewer #1: All comments have been addressed

2. Is the manuscript technically sound, and do the data support the conclusions?

Reviewer #1: Yes

3. Has the statistical analysis been performed appropriately and rigorously? 

Reviewer #1: N/A

4. Have the authors made all data underlying the findings in their manuscript fully available?

Reviewer #1: Yes

5. Is the manuscript presented in an intelligible fashion and written in standard English?

Reviewer #1: Yes

6. Review Comments to the Author

Reviewer #1: I think the authors have addressed all the comments satisfactorily and corrected the language to the extent possible from their side.

7. PLOS authors have the option to publish the peer review history of their article (what does this mean?). If published, this will include your full peer review and any attached files.

Reviewer #1: No

---

## [Editor Report · Acceptance letter]

1 Jul 2021

PONE-D-21-02576R3 

Does co-inoculation of mycorrhiza and *Piriformospora indica* fungi enhance the efficiency of Chlorophyll fluorescence and essential oil composition in peppermint under irrigation with saline water from the Caspian Sea? 

Dear Dr. Khalvandi:

I'm pleased to inform you that your manuscript has been deemed suitable for publication in PLOS ONE. Congratulations! Your manuscript is now with our production department. 

Kind regards, 

on behalf of

Dr. Mayank Gururani 

Academic Editor

PLOS ONE